# UNIVERSAL MULTI-DOMAIN TRANSLATION VIA DIFFUSION ROUTERS

**Duc Kieu**[1,*] **Kien Do**[1,*] **Tuan Hoang**[1] **Thao Minh Le**[2] **Tung Kieu**[3]

**Dang Nguyen**[1] **Thin Nguyen**[1]

[1]Applied Artificial Intelligence Intiative, Deakin University
[2]Pennsylvania State University [3]Aalborg University

[1]{v.kieu, k.do, tuan.h, d.nguyen, thin.nguyen}@deakin.edu.au
[2]mxl6224@psu.edu [3]tungkvt@cs.aau.dk

[*]Equal contribution.

## ABSTRACT

Multi-domain translation (MDT) aims to learn translations between multiple domains, yet existing approaches either require fully aligned tuples or can only handle domain pairs seen in training, limiting their practicality and excluding many cross-domain mappings. We introduce universal MDT (UMDT), a generalization of MDT that seeks to translate between any pair of $K$ domains using only $K-1$ paired datasets with a central domain. To tackle this problem, we propose Diffusion Router (DR), a unified diffusion-based framework that models all central↔non-central translations with a single noise predictor conditioned on the source and target domain labels. DR enables indirect non-central translations by routing through the central domain. We further introduce a novel scalable learning strategy with a variational-bound objective and an efficient Tweedie refinement procedure to support direct non-central mappings. Through evaluation on three large-scale UMDT benchmarks, DR achieves state-of-the-art results for both indirect and direct translations, while lowering sampling cost and unlocking novel tasks such as sketch↔segmentation. These results establish DR as a scalable and versatile framework for universal translation across multiple domains. The code is available at https://github.com/kvmduc/DiffusionRouter

## 1 INTRODUCTION

Paired domain translation, which aims to learn a mapping between two domains given aligned samples, underpins a wide range of applications, including image-to-image translation (Isola et al., 2017; Park et al., 2019), image captioning (Xu et al., 2015; Fang et al., 2015), and text-to-speech synthesis (van den Oord et al., 2016; Shen et al., 2018). Despite remarkable progress in two-domain settings, many real-world problems inherently involve multiple domains, motivating the study of *Multi-Domain Translation* (MDT).

Existing MDT approaches usually fall into two paradigms: (i) training on fully aligned tuples across domains (Wu & Goodman, 2018; Shi et al., 2019; Bao et al., 2023; Le et al., 2025), or (ii) training on multiple paired datasets with a shared central domain (Huang et al., 2022; Zhang et al., 2023; Huang et al., 2023; Koley et al., 2024). The former quickly becomes impractical as the number of domains grows due to the difficulty of collecting large-scale aligned tuples. The latter scales better but mainly supports translations between the central domain and each non-central domain, leaving cross non-central translations unaddressed.

In this paper, we introduce *Universal Multi-Domain Translation* (UMDT), which combines the ambition of enabling translations between *any* pairs of $K$ domains with the practicality of only requiring $K-1$ paired datasets involving a central domain. UMDT captures many real-world scenarios, such

as image↔text↔audio translation or multilingual machine translation, where fully aligned tuples are scarce but pairwise datasets with a pivot domain (e.g., text or English) are abundant.

To address UMDT, we propose *Diffusion Router* (DR), a novel diffusion-based framework that supports arbitrary cross-domain translations with only a single noise prediction network $\epsilon_\theta$. Inspired by network routers that determine paths using source and destination IP addresses, DR conditions $\epsilon_\theta$ on both source and target domain labels, guiding the denoising process along the correct translation path. This design avoids training a separate model for each mapping, enabling scalability to large numbers of domains. DR can perform *indirect* translation between non-central domains via the central domain. To further enable *direct* non-central translations, we introduce a scalable learning strategy that minimizes a variational upper bound on the KL divergence between indirect and parameterized direct mappings. This reduces to aligning two conditional noise predictions–one conditioned on the source non-central-domain sample and the other on its paired central-domain sample–resembling the training objectives of diffusion models. To improve efficiency, we develop Tweedie refinement, a lightweight sampling procedure that approximates conditional samples in only a few steps, greatly reducing computational cost and facilitating scalable training.

Extensive experiments and ablation studies on three newly constructed large-scale UMDT datasets demonstrate that DR consistently outperforms state-of-the-art GAN-, flow-, and diffusion-based baselines for multi-domain translations, either indirect or direct. Moreover, DR naturally generalize to more complex UMDT topologies such as spanning trees with multiple central domains.

In summary, our main contributions are:

- We formalize Universal Multi-Domain Translation (UMDT), a general setting that aims to learn translations between *any* pairs of $K$ domains using only $K-1$ paired datasets.

- We propose the Diffusion Router (DR), a unified diffusion-based framework that models all central↔non-central mappings with a single noise predictor.

- We develop a scalable learning strategy with a variational-bound objective and Tweedie refinement to enable direct non-central translations.

- We construct three new UMDT benchmarks and show that DR achieves state-of-the-art results for both indirect and direct translations.

## 2 PRELIMINARIES

### 2.1 DIFFUSION MODELS

Diffusion models (Sohl-Dickstein et al., 2015; Song & Ermon, 2019; Ho et al., 2020; Song et al., 2021a) are a generative framework that generates data from noise by reversing a forward diffusion process. The forward process is typically a Markov process with the transition kernel $p(x_t|x_{t-1})$ chosen such that the marginal distribution $p(x_t|x)$ (with $x_0 \equiv x$) is a Gaussian distribution of the form $\mathcal{N}(a_t x, \sigma_t^2 I)$. This enables direct sampling of $x_t$ from $x$ as:

$$x_t = a_t x + \sigma_t \epsilon \tag{1}$$

Here, $\epsilon \sim \mathcal{N}(0, I)$; $a_t$ and $\sigma_t$ are predefined time-dependent coefficients satisfying $1 \approx a_0 > \cdots > a_T \approx 0$ and $0 \approx \sigma_0^2 < \cdots < \sigma_T^2 \approx 1$. At the final time step, $p(x_T \mid x) \approx p(x_T) = \mathcal{N}(0, I)$.

It has been shown that (Song et al., 2021a) the reverse transition kernel $p_\theta(x_{t-1}|x_t)$ can be parameterized as a Gaussian distribution $\mathcal{N}\left(\mu_{\theta,t,t-1}(x_t), \omega_{t-1|t}^2 I\right)$ where the mean is defined as:

$$\mu_{\theta,t,t-1}(x_t) = \frac{a_{t-1}}{a_t} x_t + \left(\sqrt{\sigma_{t-1}^2 - \omega_{t-1|t}^2} - \frac{\sigma_t a_{t-1}}{a_t}\right) \epsilon_\theta(x_t, t), \tag{2}$$

and the variance is $\omega_{t-1|t}^2 = \eta^2 \sigma_{t-1}^2 \left(1 - \frac{\sigma_{t-1}^2}{\sigma_t^2} \frac{a_t^2}{a_{t-1}^2}\right)$ with $\eta \in [0, 1]$ (Song et al., 2021a).

The noise prediction network $\epsilon_\theta$ is trained to predict the noise $\epsilon$ used to construct $x_t$ in the forward process (see Eq. 1), using the following loss (Ho et al., 2020):

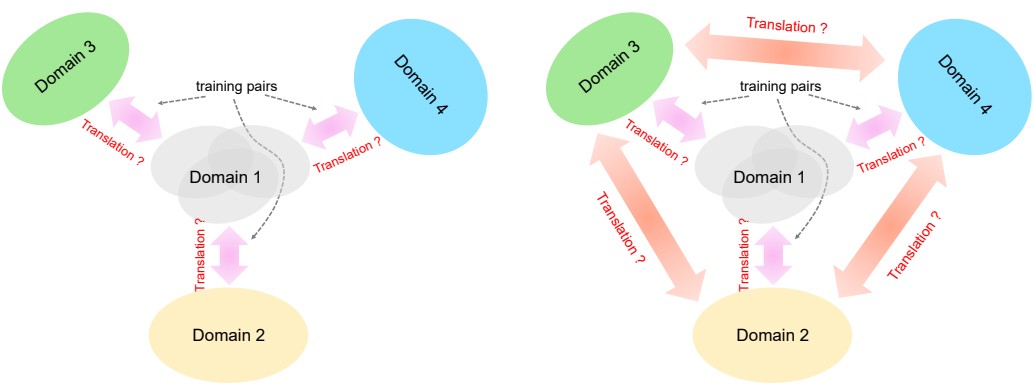

(a) Conventional multi-domain translation      (b) Universal multi-domain translation

Figure 1: Illustration of conventional and universal multi-domain translation

$$\mathcal{L}(\theta) = \mathbb{E}_{x,t,\epsilon}\left[\|\epsilon_\theta(x_t, t) - \epsilon\|_2^2\right], \tag{3}$$

where $x \sim p(x)$, $t \sim \mathcal{U}(1,T)^1$, and $\epsilon \sim \mathcal{N}(0, \mathrm{I})$. This objective can be interpreted as a variational upper bound on $-\log p(x)$ (Sohl-Dickstein et al., 2015; Ho et al., 2020; Song et al., 2021b).

While diffusion models were initially proposed to model unconditional data distributions $p(x)$, they can be extended to conditional distributions $p(x|y)$ (Dhariwal & Nichol, 2021; Ho & Salimans, 2021; Rombach et al., 2022), by incorporating the condition $y$ into the noise predictor, i.e., $\epsilon_\theta(x_t, t, y)$. This conditioning can be implemented either by concatenating $y$ with $x_t$ (Saharia et al., 2022), or through cross attention between $x_t$ and $y$ (Rombach et al., 2022).

## 3 UNIVERSAL MULTI-DOMAIN TRANSLATION

We study a more general and challenging extension of conventional multi-domain translation (MDT) problem (Fig. 1a), which we term *universal multi-domain translation* (UMDT) (Fig. 1b). In this setting, we consider $K$ distinct domains, $X^1, X^2, ..., X^K$, with training data consisting of $K-1$ paired datasets between each domain $X^k$ and a *shared central domain* $X^c$ (where $1 \le c \le K$): $\mathcal{D}_{k,c} = \left\{\left(x^k, x^c\right)\right\}_{n=1}^{N_k}$, for all $k \ne c$. The goal is to learn a model that can translate between *any* pair of domains $\left(X^i, X^j\right)$ for *all* $i \ne j$. Since paired data between non-central domains are unavailable, we assume that samples from the central domain $X^c$ share overlapping information across the paired datasets. This is a mild assumption and is typically satisfied in practice.

UMDT is highly practical. For example, in multi-modal translation across images, text, and audio, it is often difficult to obtain large-scale datasets with fully aligned triplets (image, text, audio). However, paired datasets such as image-text (e.g., image captions) and text-audio (e.g., audiobooks) are more common. In this case, text naturally serve as the central domain, with image and audio as non-central domains. Image↔audio translation can then be achieved indirectly via text, even without direct image-audio training pairs. Importantly, text samples across datasets *need not match exactly*; loose overlaps–such as shared vocabulary or semantics–are sufficient for image↔audio translation.

More generally, UMDT can be extended to cases where the $K-1$ paired datasets form a *spanning tree* over the $K$ domains. Since spanning trees can take arbitrary structures, in this work we restrict our attention to the star-shaped configuration. Nonetheless, the method proposed in the following section naturally generalizes to the broader spanning-tree setting.

---

[1]$\mathcal{U}(1,T)$ denotes a uniform distribution of time steps between 1 and $T$.

## 4 DIFFUSION ROUTERS

### 4.1 INDIRECT TRANSLATION BETWEEN NON-CENTRAL DOMAINS VIA THE CENTRAL ONE

From a probabilistic perspective, bidirectional translation between two non-central domains $X^i$ and $X^j$ can be viewed as sampling from the conditional distributions $p\left(x^i|x^j\right)$ and $p\left(x^j|x^i\right)$. These can be expressed via the central domain $X^c$ as:

$$p\left(x^j|x^i\right) = \int p\left(x^j|x^c\right) p\left(x^c|x^i\right) dx^c, \quad p\left(x^i|x^j\right) = \int p\left(x^i|x^c\right) p\left(x^c|x^j\right) dx^c \qquad (4)$$

Here, we assume $X^i \perp X^j | X^c$ for all $i,j \neq c$ which leads to $p\left(x^i|x^c\right) = p\left(x^i|x^c, x^j\right)$ and $p\left(x^j|x^c\right) = p\left(x^j|x^c, x^i\right)$. In other words, once the central domain $X^c$ is known, the non-central domains become conditionally independent of each other.

This formulation extends the transitivity property of equivalence relations into a probabilistic framework. It implies that if we can learn bidirectional mappings between the central domain $X^c$ and each non-central domain $X^k$ ($k \neq c$) that capture the couplings $p\left(x^k|x^c\right)$ and $p\left(x^c|x^k\right)$ from the training data (i.e., solving the conventional MDT problem), then we can perform indirect translation between any pair $\left(X^i, X^j\right)$ through $X^c$, even without direct supervision between $X^i$ and $X^j$.

Notably, $p\left(x^k|x^c\right)$ and $p\left(x^c|x^k\right)$ can be effectively modeled using conditional diffusion models as discussed in Section 2 or diffusion bridges (Liu et al., 2023a; Li et al., 2023; Zhou et al., 2024; Kieu et al., 2025) (with details in Appdx. B.1). However, naively constructing a separate model for each distribution would require up to $2(K-1)$ models to cover all translations between $X^c$ and the non-central domains $\left\{X^k|k \neq c\right\}$, which becomes impractical as the number of domains grows.

To address this scalability challenge, we propose a unified framework called *Diffusion Router* (DR) that learns all *bidirectional* mappings between the central and non-central domains using a *single* network, thereby avoiding redundancy and enabling efficient multi-domain translation.

Inspired by network routers, which rely on source and destination IP addresses to determine the routing path, our framework incorporates the labels (or indices) of the source and target domains into the network $\epsilon_\theta$. This design allows $\epsilon_\theta$ to infer the correct translation path for a given noisy input $x_t$. Specifically, $\epsilon_\theta$ takes the form $\epsilon_\theta\left(x_t^{\text{tgt}}, t, x^{\text{src}}, \text{tgt}, \text{src}\right)$ where src and tgt denote the source and target domain labels. We train DR with the following objective function:

$$\mathcal{L}_{\text{paired}}\left(\theta\right) = \mathbb{E}_{(x^k,x^c)\sim\mathcal{D}_{k,c},t,\epsilon,\zeta}\left[\zeta\left\|\epsilon_\theta\left(x_t^k, t, x^c, k, c\right) - \epsilon\right\|_2^2 + (1-\zeta)\left\|\epsilon_\theta\left(x_t^c, t, x^k, c, k\right) - \epsilon\right\|_2^2\right],$$
$$(5)$$

where $t \sim \mathcal{U}\left(1, T\right)$, $\epsilon \sim \mathcal{N}\left(0, \text{I}\right)$, and $\zeta \sim \mathcal{B}\left(0.5\right)^2$. Next, $x_t^k = a_t x^k + \sigma_t \epsilon$ for the *standard* (or diffusion-based) DR variant and $x_t^k = \alpha_t x^k + \beta_t x^c + \sigma_t \epsilon$ for the *bridge-based* DR variant.

Once trained, DR can translate between two non-central domains $X^i$ and $X^j$ indirectly via $X^c$, using a *two-stage* process: 1) generate central-domain samples $x^c$ conditioned on a source sample $x^i$ (or $x^j$), then 2) generate target samples $x^j$ (or $x^i$) conditioned on the intermediate samples $x^c$.

### 4.2 DIRECT TRANSLATION BETWEEN NON-CENTRAL DOMAINS

Indirect translation requires generating intermediate samples $x^c$ of the central domain, which is computationally expensive and sensitive to sample quality. To overcome these drawbacks, we propose a novel approach that enables *direct* translation between non-central domains $X^i$ and $X^j$ by explicitly modeling $p_\theta\left(x^j|x^i\right)$ and $p_\theta\left(x^i|x^j\right)$. Our method can either finetune a DR pretrained with Eq. 5 or train a new DR from scratch for direct cross-domain translation.

To learn $p_\theta\left(x^j|x^i\right)$ (or similarly $p_\theta\left(x^i|x^j\right)$), we minimize the following KL divergence:

$$\mathbb{E}_{p(x^i)}\left[D_{KL}\left[p\left(x^j|x^i\right)\|p_\theta\left(x^j|x^i\right)\right]\right]$$
$$= \mathbb{E}_{(x^i,x^c)\sim\mathcal{D}_{i,c}}\mathbb{E}_{p(x^j|x^c)}\left[\log\left(\mathbb{E}_{p(x'^c|x^i)}\left[p\left(x^j|x'^c\right)\right]\right) - \log p_\theta\left(x^j|x^i\right)\right], \qquad (6)$$

---

$^2\mathcal{B}\left(0.5\right)$ denotes a Bernoulli distribution with the probability of getting 1 equal 0.5.

where $x'^c$ denotes samples of $X^c$ that are distinct from $x^c$. The detailed derivation of Eq. 6 is provided in Appdx. A.1. The main bottleneck here is the term $\log\left(\mathbb{E}_{p(x'^c|x^i)}\left[p\left(x^j|x'^c\right)\right]\right)$. First, sampling from $p\left(x'^c|x^i\right)$ typically requires hundreds to thousands of denoising steps if using a pretrained DR. Second, even if we obtain samples $x'^c$ from $p\left(x'^c|x^i\right)$, evaluating $p\left(x^j|x'^c\right)$ remains intractable due to the lack of a closed-form expression.

To overcome this, we approximate $\mathbb{E}_{p(x'^c|x^i)}\left[p\left(x^j|x'^c\right)\right]$ by $p\left(x^j|x^c\right)$ where $x^c$ comes from the pair $\left(x^i, x^c\right) \sim \mathcal{D}_{i,c}$. This leads to the following tractable training objective:

$$\mathbb{E}_{(x^i,x^c)\sim\mathcal{D}_{i,c}}\mathbb{E}_{p(x^j|x^c)}\left[\log\left(\mathbb{E}_{p(x'^c|x^i)}\left[p\left(x^j|x'^c\right)\right]\right) - \log p_\theta\left(x^j|x^i\right)\right]$$

$$\approx \mathbb{E}_{(x^i,x^c)\sim\mathcal{D}_{i,c}}\mathbb{E}_{p(x^j|x^c)}\left[\log p\left(x^j|x^c\right) - \log p_\theta\left(x^j|x^i\right)\right] \tag{7}$$

$$= \mathbb{E}_{(x^i,x^c)\sim\mathcal{D}_{i,c}}\left[D_{KL}\left(p\left(x^j|x^c\right)\|p_\theta\left(x^j|x^i\right)\right)\right] \tag{8}$$

Although this approximation introduces bias due to the logarithm operating on a (single-sample) Monte Carlo estimate, we empirically observe that its impact on learning is manageable, particularly when the conditional distribution $p\left(x'^c|x^i\right)$ is sharply peaked at its mode.

The objective in Eq. 8 suggests that we can learn a direct mapping from $X^i$ to $X^j$ by ensuring that if $x^i$ and $x^c$ are semantically aligned (i.e., appear as a pair in $\mathcal{D}_{i,c}$), then the conditional distribution over the target domain $X^j$ given $x^i$ should closely match that given $x^c$. Note that $p\left(x^j|x^c\right)$ can be viewed as the path distribution of the stochastic process $X^c \to X^j$, modeled using a *pretrained* DR introduced in Section 4.1. Likewise, $p_\theta\left(x^j|x^i\right)$ represents the path distribution for the process $X^i \to X^j$, which we also model using the same DR. This is implemented by providing $i, j$ as inputs to the noise prediction network $\epsilon_\theta$ of the DR. As a result, learning $p_\theta\left(x^j|x^i\right)$ corresponds to finetuning this pretrained DR to support direct mapping from $X^i$ to $X^j$.

Let $p_{\text{ref}}\left(x^j|x^c\right)$ denote the parameterization of $p\left(x^j|x^c\right)$ given by the pretrained DR. Instead of minimizing the KL divergence between $p_{\text{ref}}\left(x^j|x^c\right)$ and $p_\theta\left(x^j|x^i\right)$ directly, we minimize the sum of KL divergences between their respective transition kernels. This sum acts as a variational upper bound on the original KL objective, as shown in Appdx. A.2:

$$\mathbb{E}_{(x^i,x^c)\sim\mathcal{D}_{i,c}}\left[D_{KL}\left(p_{\text{ref}}\left(x^j|x^c\right)\|p_\theta\left(x^j|x^i\right)\right)\right]$$

$$\leq \mathbb{E}_{(x^i,x^c)\sim\mathcal{D}_{i,c}}\left[\sum_{t=1}^{T}\mathbb{E}_{p_{\text{ref}}\left(x_t^j|x^c\right)}\left[D_{KL}\left(p_{\text{ref}}\left(x_{t-1}^j|x_t^j,x^c\right)\|p_\theta\left(x_{t-1}^j|x_t^j,x^i\right)\right)\right]\right] + \text{const} \tag{9}$$

Ideally, the sampling distribution in Eq. 9 should be $p_{\text{ref}}\left(x_t^j|x^i\right)$ so that after training, $p_\theta\left(x_{t-1}^j|x_t^j,x^i\right)$ can accurately model the transition dynamics from $X^i$ to $X^j$. However, the actual sampling distribution is $p_{\text{ref}}\left(x_t^j|x^c\right)$. For bridge-based DR, the stochastic processes $X^i \to X^j$ and $X^c \to X^j$ starts from different initial states $x^i$ and $x^c$, respectively, making their path distributions, $p_{\text{ref}}\left(x_t^j|x^i\right)$ and $p_{\text{ref}}\left(x_t^j|x^c\right)$ inherently different. Therefore, bridge-based DR are ill-suited for finetuning with the objective in Eq. 9. By contrast, standard DR provide a more viable solution. Since both $X^i \to X^j$ and $X^c \to X^j$ originate from the same Gaussian prior, they share a common stochastic path, allowing $p_{\text{ref}}\left(x_t^j|x^c\right)$ to serve as a proxy for $p_{\text{ref}}\left(x_t^j|x^i\right)$. For this reason, we restrict our focus to finetuning standard DR.

By applying the standard reparameterization trick for diffusion models (Ho et al., 2020), we reformulate Eq. 9 as a noise prediction loss:

$$\mathcal{L}_{\text{unpaired}}\left(\theta\right) = \mathbb{E}_{(x^i,x^c)\sim\mathcal{D}_{i,c},x_t^j\sim p_{\text{ref}}\left(x_t^j|x^c\right),t,\epsilon}\left[\left\|\epsilon_\theta\left(x_t^j,t,x^i,j,i\right) - \epsilon_{\text{ref}}\left(x_t^j,t,x^c,j,c\right)\right\|_2^2\right], \tag{10}$$

where $\epsilon_{\text{ref}}$ is the frozen noise prediction network of the pretrained DR.

To prevent catastrophic forgetting of previously learned mappings between $X^c$ and $X^k$, we combine Eq. 10 with $\mathcal{L}_{\text{paired}}$ (see Eq. 5) resulting in the final loss:

$$\mathcal{L}_{\text{final}}\left(\theta\right) = \lambda_1\mathcal{L}_{\text{unpaired}}\left(\theta\right) + \lambda_2\mathcal{L}_{\text{paired}}\left(\theta\right) \tag{11}$$

Figure 2: Tweedie refinement with $n \in \{0, 1, 3, 5, 7\}$ on Faces-UMDT-Latent. **Left**: A conditional sample $x^c$ and a random target-domain sample $x^j$. **Middle**: A ground-truth noisy target-domain sample $x_t^j$ aligned with $x^c$ (*not* available during training). **Right**: Tweedie refinement progressively transforms $x_t^j \sim p\left(x_t^j\right)$ into $x_t^j \sim p\left(x_t^j | x^c\right)$ as $n$ increases.

Here, the coefficients $\lambda_1, \lambda_2 \geq 0$ balance the trade-off between learning new translations $X^i \rightarrow X^j$ and preserving existing mappings $X^k \leftrightarrow X^c$.

Notably, $\mathcal{L}_{final}(\theta)$ is highly flexible. It can train DR *from scratch* by treating $\epsilon_{ref}$ as an online network with frozen parameters rather than a pretrained model (Appdx. D.3.3). Moreover, it supports in the case where paired domains form a spanning tree with multiple central domains (see Section 5).

### 4.2.1 SAMPLING FROM CONDITIONAL DISTRIBUTIONS WITH TWEEDIE REFINEMENT

The main challenge in Eq. 10 is sampling $x_t^j$ from $p_{ref}\left(x_t^j | x^c\right)$. A straightforward approach is to perform backward denoising from time $T$ to $t$ using the pretrained DR, but this is computationally expensive and does not scale well. To address this, we propose a novel sampling method:

$$x_{t,(n+1)}^j = x_{t,(n)}^j + \sigma_t \left(\epsilon - \epsilon_\theta \left(x_{t,(n)}^j, t, x^c, j, c\right)\right) \tag{12}$$

where $\epsilon \sim \mathcal{N}(0, \mathrm{I})$ and $x_{t,(n)}^j$ denotes the refined sample after $n$ steps, initialized with $x_{t,(0)}^j \sim p_{ref}\left(x_t^j\right)$. A sample $x_t^j \sim p_{ref}\left(x_t^j\right)$ can be obtained by first drawing $\left(x^j, \cdot\right)$ from $\mathcal{D}_{j,c}$ and then applying the forward diffusion process $x_t^j = a_t x^j + \sigma_t \epsilon$.

We refer to this procedure as *Tweedie refinement* due to its connection with Tweedie's formula (Efron, 2011). Empirically, we find that Tweedie refinement can approximate samples from $p_{ref}\left(x_t^j | x^c\right)$ with only a few refinement steps (see Fig. 2). Compared with existing refinement techniques (Song et al., 2021b; Yu et al., 2023), our approach (1) introduces a distinct formulation, (2) converts unconditional samples into conditional ones rather than projecting off-distribution samples back onto a marginal distribution, and (3) is applied during training rather than inference.

## 5 EXPERIMENT

### 5.1 EXPERIMENTAL SETUP

#### 5.1.1 DATASETS

Since the proposed UMDT problem is novel, no datasets currently exist for it. To address this, we create three benchmark datasets for evaluating our method, namely **Shoes-UMDT**, **Faces-UMDT**, and **COCO-UMDT**. Detailed descriptions of these datasets are provided below.

**Shoes-UMDT**  This dataset is adapted from the Edges2Shoes dataset (Isola et al., 2017) used for paired image-to-image translation. From the original 50K (shoe, edge) pairs, we randomly sample two *disjoint* subsets of 20K pairs each. One subset remains unchanged, while in the other, the "edge" image in each pair is replaced with a "grayscale" version of the corresponding "shoe" image, rotated by 20 degrees and scaled by 20% smaller . This yields two disjoint sets: 20K (shoe, edge) pairs and 20K (shoe, grayscale) pairs. In this setup, "shoe" is the central domain, while "edge" and "grayscale" are non-central domains. The remaining 10K (shoe, edge) pairs are used to generate

| Method | FID↓ | | | | | |
| --- | --- | --- | --- | --- | --- | --- |
| | Shoes-UMDT | | | Faces-UMDT-Latent | | |
| | Edge↔Shoe | Gray.↔Shoe | Edge↔Gray. | Ske.↔Face | Seg.↔Face | Ske.↔Seg. |
| StarGAN | 9.92/20.18 | 19.73/42.61 | 18.64/27.41 | - | - | - |
| Rectified Flow | 2.88/30.92 | 3.75/43.38 | 20.14/18.83 | 20.22/97.76 | 10.85/81.44 | 50.82/17.31 |
| UniDiffuser | 2.98/11.94 | 2.72/4.40 | 4.81/12.26 | 13.13/55.46 | 11.02/46.04 | 36.13/12.52 |
| iDR | **1.66/5.15** | **0.53/1.60** | **1.85/5.48** | **9.07/23.88** | 6.12/**19.12** | **15.37**/6.15 |
| dDR | 2.01/5.76 | 0.57/1.69 | 2.74/6.51 | 9.62/27.09 | **3.43**/21.26 | 19.42/**5.52** |

Table 1: FID scores on Shoes-UMDT and Faces-UMDT-Latent. Translations without paired data are marked in brown. The best results are shown in **bold**, and the second-best are underlined.

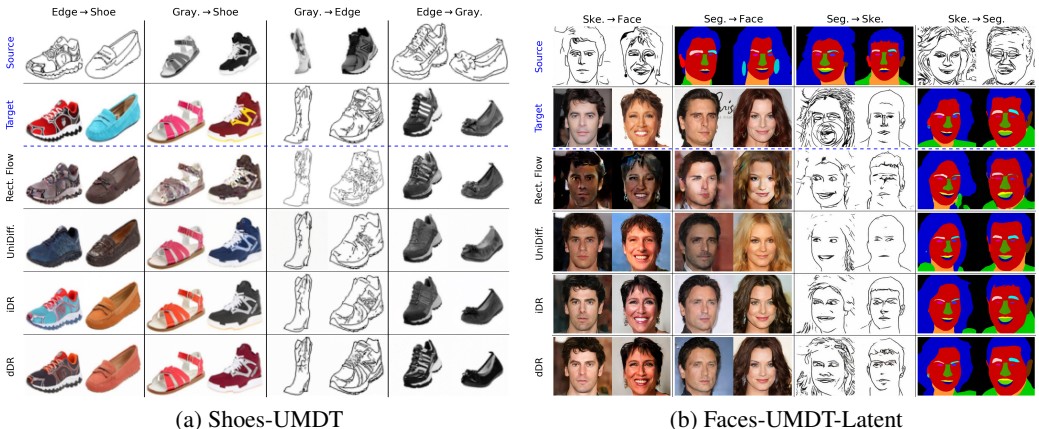

(a) Shoes-UMDT                                    (b) Faces-UMDT-Latent

Figure 3: Qualitative results on Shoes-UMDT and Faces-UMDT-Latent.

grayscale images following the same procedure as in the training data, producing a test set of 10K (shoe, edge, grayscale) triplets. All images are resized to 64×64×3.

**Faces-UMDT**  We build Faces-UMDT by combining CelebA-Mask-HQ (Lee et al., 2020) (30K (face, segment) pairs) and FFHQ (Karras et al., 2019) (70K face images). From FFHQ, we randomly select 30K face images and use Sobel filter followed by sketch simplifier (Simo-Serra et al., 2018) to generate corresponding sketches, producing 30K (face, sketch) pairs. These are merged with 25K randomly selected (face, segment) pairs from CelebA-Mask-HQ to form the training set in which "face" is the central domain and "segment", "sketch" are non-central domains. For testing, we generate a sketch for each face image in the remaining 5K CelebA-Mask-HQ pairs, resulting in 5K (face, segment, sketch) triplets.

This setup reflects real-world scenarios where the two disjoint subsets of face images associated with CelebA-Mask-HQ and FFHQ follow distinct distributions, making sketch↔segment translation via the face domain more challenging. For this dataset, we consider two settings: 1) resizing input images to 128×128×3 and translating in the pixel space, and 2) encoding 256×256×3 images using a VAE encoder (Rombach et al., 2022) and translating in the latent space of shape 32×32×4. These settings result in two versions which are Faces-UMDT-Pixel and Face-UMDT-Latent, respectively.

**COCO-UMDT-Star and COCO-UMDT-Chain**  COCO-Stuff (Caesar et al., 2018) is a large and diverse image segmentation dataset with 118K (color, segment) pairs for training and 5K pairs for evaluation, covering 80 "thing" classes, 91 "stuff" classes, and one "unlabeled" class. Following (Mou et al., 2024), we generate additional domains by applying the Pixel Difference Network (Su et al., 2021) to extract sketches and MiDaS (Ranftl et al., 2020) to produce depth maps from all color images. For COCO-UMDT-Star, we construct three training subsets of 70K color images each,

| Method | FID↓ | | | | | |
| --- | --- | --- | --- | --- | --- | --- |
| | Ske.↔Color | Seg.↔Color | Depth↔Color | Ske.↔Seg. | Ske.↔Depth | Seg.↔Depth |
| Rectified Flow | 23.18/80.80 | 54.00/142.15 | 17.32/112.64 | 64.47/75.58 | 78.41/28.69 | 79.20/35.53 |
| UniDiffuser | 15.39/40.93 | 35.81/89.58 | 12.64/59.72 | 39.62/38.44 | 28.12/15.72 | 38.39/23.41 |
| iDR | 10.72/21.73 | 21.64/29.28 | 7.25/24.19 | **22.77/22.96** | **17.88/8.63** | **23.19/12.00** |
| dDR | **10.12/20.94** | **21.23/28.32** | **7.00/23.20** | 26.73/23.64 | 20.75/9.42 | 24.91/14.87 |

Table 2: FID scores on COCO-UMDT-Star. Translations without paired data are marked in brown. The best results are shown in **bold**, and the second-best are underlined.

| Method | FID↓ | | | | | |
| --- | --- | --- | --- | --- | --- | --- |
| | Ske.↔Color | Seg.↔Color | Depth↔Color | Ske.↔Seg. | Ske.↔Depth | Seg.↔Depth |
| Rectified Flow | 22.33/85.03 | 57.53/148.57 | 27.49/146.27 | 65.18/79.91 | 68.27/21.02 | 94.55/42.40 |
| UniDiffuser | 15.43/45.51 | 36.89/92.38 | 14.40/68.64 | 39.97/38.70 | 26.49/12.40 | 39.85/24.51 |
| iDR | **10.47/26.73** | **23.14**/38.60 | **8.55/39.66** | **26.52/25.02** | **14.73/7.54** | **26.08/14.15** |
| dDR | 11.11/28.39 | 24.68/**38.47** | 10.69/44.23 | 28.13/26.94 | 14.92/7.82 | 30.98/18.61 |

Table 3: FID scores on COCO-UMDT-Chain. Translations without paired data are marked in brown. The best results are shown in **bold**, and the second-best are underlined.

paired with segmentation maps, sketches, and depth maps, respectively, yielding (color, segment), (color, sketch), and (color, depth) training pairs. For COCO-UMDT-Chain, we form three paired subsets: (segment, color), (color, sketch), and (sketch, depth), each with 70K pairs. Both COCO-UMDT-Star and COCO-UMDT-Chain share a common test set of 5K (color, segment, sketch, depth) quadruplets derived from the original evaluation split. All images are resized to $256\times256\times3$ and mapped into latent space of size $32\times32\times4$ using VAE encoder from (Rombach et al., 2022).

### 5.1.2 Baselines and metrics

We consider two versions of DR. The first is trained with the loss $\mathcal{L}_{\text{paired}}(\theta)$ in Eq. 5, which can only perform indirect translations between non-central domains. The second is finetuned from the first using the loss $\mathcal{L}_{\text{final}}(\theta)$ in Eq. 11, allowing direct cross-domain translation. We refer to these two versions as iDR and dDR, respectively. We also train version from scratch using $\mathcal{L}_{\text{final}}(\theta)$ and compare with the finetuned version in Appdx. D.3.3.

As with datasets, methods for addressing the UMDT problem remain largely unexplored. To establish baselines, we adapt several approaches originally designed for the conventional MDT setting. These include StarGAN (Choi et al., 2018), UniDiffuser (Bao et al., 2023), and Rectified Flow (Liu et al., 2022) as representatives for GAN-based, diffusion-based, and flow-based methods. Further implementation details for both our method and the baselines are provided in Appdx. D.1.

Following previous works on MDT, we use FID (Heusel et al., 2017) and LPIPS (Zhang et al., 2018) to measure distributional fidelity and perceptual similarity, respectively. In all tables, each entry for translations A↔B is reported as X/Y, where X is the metric for A←B and Y is the metric for A→B.

### 5.2 Results

We report the quantitative results for Shoes-UMDT and Faces-UMDT-Latent in Table 1, for COCO-UMDT-Star in Table 2, and for Faces-UMDT-Pixel in Table 5 of Appdx D.2.2. iDR consistently outperforms all baselines by a significant margin across all benchmarks, highlighting the effectiveness of conditioning $\epsilon_\theta$ on both source and target domain labels to guide translation. The weak performance of StarGAN in central↔non-central translations ($X^c \leftrightarrow X^k$) can be attributed to two main factors: (i) its relatively outdated generator architecture, and (ii) its original design for unpaired rather than paired translation. Despite being trained on aligned $(x^c, x^k)$ pairs, Rectified Flow (RF) and UniDiffuser struggle to learn robust $X^c \leftrightarrow X^k$ mappings. For RF, performance degrades most

noticeably when the target domain is diverse and high-variance (e.g., Edge→Shoe or Seg.→Face), as its deterministic formulation cannot capture the stochasticity of $p\left(x^c|x^k\right)$ and $p\left(x^k|x^c\right)$. For UniDiffuser, the repeated substitution of missing domains with Gaussian noise during training and inference likely undermines its ability to accurately modeling the joint distribution over all domains.

dDR shows slight decreases in performance for translation tasks without paired data compared to iDR, yet still outperforms all baselines significantly. The performance drop can be attributed to imperfections in our refinement procedure and to bias introduced by the Monte Carlo approximation inside the logarithm in Eq. 7. Importantly, dDR reduces the number of sampling steps for non-central translations by half relative to iDR–a substantial efficiency gain, given that iDR typically requires hundreds to thousands of steps for cross-domain translation. As shown in Fig. 3, iDR and dDR produce higher-quality samples compared to the baselines.

We further observe consistent behavior on COCO-UMDT-Chain in Table 3 compared to COCO-UMDT-Star: dDR improves efficiency by 2-3 times for translations without paired data while incurring only marginal performance degradation compared to iDR. This empirically validate the generalizability of our learning strategy for UMDT beyond star-shaped structures.

## 5.3 ABLATION STUDIES

Owing to space limitations, we conduct comprehensive ablations to quantify the effect of key hyperparameter choices and report at Appdx. D.3. Specifically, we (i) evaluate Tweedie refinement by varying the number of refinement steps $n \in \{0, 1, 3, 5\}$ (Appdx. D.3.1); (ii) study the effect of the rehearsal coefficient $\lambda_2 \in \{0, 0.3, 1, 3\}$ in the loss $\mathcal{L}_{\text{final}}$ during finetuning (Appdx. D.3.2); compare dDR trained from scratch versus finetuned from a pretrained iDR (Appdx. D.3.3); (iv) compare the performances of iDR and dDR across different numbers of sampling steps (Appdx. D.3.4); and (v) benchmark standard DR against several bridge-based DR variants on UMDT (Appdx. D.3.5).

## 6 RELATED WORK

Due to space constraints, we primarily review diffusion-based approaches for multi-domain translation (MDT), with other methods discussed in Appdx. C. Building on the success of Stable Diffusion (SD) (Rombach et al., 2022) in text-to-image generation, many works adapt SD to MDT by finetuning it to condition on additional modalities such as edges, segmentation maps, depth, or poses (Zhang et al., 2023; Huang et al., 2023; Mou et al., 2024). Versatile Diffusion (Xu et al., 2023) extends this idea with a modular cross-modal architecture, but its model size scales linearly with the number of domains and experiments are limited to text and image. CoDi (Tang et al., 2023) adopts a two-stage framework: first, modality-specific encoders are trained to align samples into a shared latent space; second, separate latent diffusion models are trained for each modality conditioned on encoder outputs and noisy paired samples. While this supports any-to-any generation, it requires contrastive pretraining and multiple diffusion models that scale linearly with domain count. UniDiffuser (Bao et al., 2023) instead models the joint distribution of all domains using a single transformer-based noise prediction network that treats domain samples as tokens, with independent noise schedules per domain. This design supports arbitrary modalities but requires fully aligned tuples across domains and long training times. Moreover, UniDiffuser must process all domains jointly, making generation costly when only a subset of domains is desired. One Diffusion (Le et al., 2025) follows a similar design but replaces the noise prediction network with a velocity network trained via flow matching (Liu et al., 2022) and introduces a different transformer architecture (Zhuo et al., 2024). Like UniDiffuser, it depends on fully aligned tuples where samples of missing modalities are synthesized. OmniFlow (Li et al., 2025) is another flow-based model for multi-modal generation like One-Diffusion. However, instead of training from scratch, it extends from SD3 (Esser et al., 2024).

In summary, most diffusion-based MDT methods either rely on contrastive learning or synthetic fully aligned data, suffer from linear growth in model size, or require processing all domains simultaneously. By contrast, our method trains directly from domain pairs, avoids model-size scaling with domain count, and flexibly adjusts sampling cost to the desired number of output domains. Although not the primary focus of this work, Diffusion Routers also enable generation of a single target domain from multiple source domains by combining the scores of models conditioned on these sources. This capability makes our method adaptable for any-to-any generation.

## 7 CONCLUSION

We introduced the universal multi-domain translation (UMDT) problem, which seeks to learn mappings between any pair of $K$ domains using only $K - 1$ paired datasets with a central domain. The main challenge lies in learning non-central↔non-central translations (NNTs), where training pairs are unavailable. To tackle this, we proposed Diffusion Router (DR), which supports both indirect NNTs through the central domain and direct NNTs. The direct variant can be obtained either by fine-tuning from the indirect version or by training from scratch. We introduced novel variational-bound objective and conditional sampling method for learning the direct variant. Empirical evaluations on three newly constructed UMDT datasets demonstrated that our method consistently outperforms existing baselines. In future work, we aim to extend DR to large-scale multimodal generation across image, text, and audio, particularly addressing scenarios where paired datasets (e.g., image↔audio) are scarce in practice.

## LLM USAGE

Large Language Models were not involved in the design of our approach. We used them solely to improve the manuscript's readability (grammar and style); none of these uses influenced the method, training, or reported results.

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

## A  THEORETICAL RESULTS

### A.1  DERIVATION OF EQ. 6

The detailed derivation of Eq. 6 is given below:

$$\mathbb{E}_{p(x^i)}\left[D_{KL}\left(p\left(x^j|x^i\right)||p_\theta\left(x^j\mid x^i\right)\right)\right]$$

$$= \int p\left(x^i\right)\left[\int p\left(x^j|x^i\right)\left(\log p\left(x^j|x^i\right) - \log p_\theta\left(x^j|x^i\right)\right)dx^j\right]dx^i \tag{13}$$

$$= \int p\left(x^i\right)\int\left(\int p\left(x^j|x^c\right)p\left(x^c|x^i\right)dx^c\right)\left(\log p\left(x^j|x^i\right) - \log p_\theta\left(x^j|x^i\right)\right)dx^j dx^i \tag{14}$$

$$= \int\int\int p\left(x^i\right)p\left(x^c|x^i\right)p\left(x^j|x^c\right)\left(\log p\left(x^j|x^i\right) - \log p_\theta\left(x^j|x^i\right)\right)dx^j dx^c dx^i \tag{15}$$

$$= \int p\left(x^i,x^c\right)\int p\left(x^j|x^c\right)\left(\log p\left(x^j|x^i\right) - \log p_\theta\left(x^j|x^i\right)\right)dx^j dx^c dx^i \tag{16}$$

$$= \mathbb{E}_{p(x^i,x^c)}\mathbb{E}_{p(x^j|x^c)}\left[\log p\left(x^j|x^i\right) - \log p_\theta\left(x^j|x^i\right)\right] \tag{17}$$

$$= \mathbb{E}_{p(x^i,x^c)}\mathbb{E}_{p(x^j|x^c)}\left[\log \mathbb{E}_{p(x'^c|x^i)}\left[p\left(x^j|x'^c\right)\right] - \log p_\theta\left(x^j|x^i\right)\right] \tag{18}$$

Note that in our derivation, $p\left(x^j|x^i\right)$ is substituted with its expression in described in Eq. 4.

### A.2  DERIVATION OF THE VARIATIONAL UPPER BOUND IN EQ. 9

The detailed derivation of the variational upper bound in Eq. 9 is as follows:

$$\mathbb{E}_{(x^i,x^c)\sim\mathcal{D}_{i,c}}\left[D_{KL}\left(p_{\text{ref}}\left(x^j|x^c\right)||p_\theta\left(x^j|x^i\right)\right)\right]$$

$$\leq \mathbb{E}_{(x^i,x^c)\sim\mathcal{D}_{i,c}}\left[D_{KL}\left(p_{\text{ref}}\left(x^j_{0:T}|x^c\right)||p_\theta\left(x^j_{0:T}|x^i\right)\right)\right] \tag{19}$$

$$= \mathbb{E}_{(x^i,x^c)\sim\mathcal{D}_{i,c}}\left[\int p_{\text{ref}}\left(x^j_{0:T}|x^c\right)\log\left(\frac{p_{\text{ref}}\left(x^j_{0:T}|x^c\right)}{p_\theta\left(x^j_{0:T}|x^i\right)}\right)dx^j_{0:T}\right] \tag{20}$$

$$= \mathbb{E}_{(x^i,x^c)\sim\mathcal{D}_{i,c}}\left[\int p_{\text{ref}}\left(x^j_{0:T}|x^c\right)\log\left(\frac{p\left(x^j_T|x^c\right)\prod_{t=1}^T p_{\text{ref}}\left(x^j_{t-1}|x^j_t,x^c\right)}{p\left(x^j_T|x^i\right)\prod_{t=1}^T p_\theta\left(x^j_{t-1}|x^j_t,x^i\right)}\right)dx^j_{0:T}\right] \tag{21}$$

$$= \mathbb{E}_{(x^i,x^c)\sim\mathcal{D}_{i,c}}\left[\mathbb{E}_{p_{\text{ref}}\left(x^j_{0:T}|x^c\right)}\left[\sum_{t=1}^T\log\left(\frac{p_{\text{ref}}\left(x^j_{t-1}|x^j_t,x^c\right)}{p_\theta\left(x^j_{t-1}|x^j_t,x^i\right)}\right)\right]\right] + \text{const} \tag{22}$$

$$= \mathbb{E}_{(x^i,x^c)\sim\mathcal{D}_{i,c}}\left[\sum_{t=1}^T\mathbb{E}_{p_{\text{ref}}\left(x^j_t|x^c\right)}\mathbb{E}_{p_{\text{ref}}\left(x^j_{t-1}|x^j_t,x^c\right)}\left[\log\left(\frac{p_{\text{ref}}\left(x^j_{t-1}|x^j_t,x^c\right)}{p_\theta\left(x^j_{t-1}|x^j_t,x^i\right)}\right)\right]\right] + \text{const} \tag{23}$$

$$= \mathbb{E}_{(x^i,x^c)\sim\mathcal{D}_{i,c}}\left[\sum_{t=1}^T\mathbb{E}_{p_{\text{ref}}\left(x^j_t|x^c\right)}D_{KL}\left(p_{\text{ref}}\left(x^j_{t-1}|x^j_t,x^c\right)||p_\theta\left(x^j_{t-1}|x^j_t,x^i\right)\right)\right] + \text{const} \tag{24}$$

Here, the inequality in Eq. 19 is a variant of the data processing inequality.

## B  ADDITIONAL PRELIMINARIES

### B.1  DIFFUSION BRIDGES

Diffusion bridges (Liu et al., 2023a; Albergo et al., 2023; Liu et al., 2023b; Li et al., 2023; Zhou et al., 2024; Kieu et al., 2025; Nguyen et al., 2025) offer an alternative approach to modeling conditional distributions $p\left(x|y\right)$ where the generation process starts from the observation $y$ rather than from

standard Gaussian noise, as in traditional diffusion models. Specifically, the corruption process is characterized by the marginal distribution $p(x_t|x, y)$, representing a stochastic trajectory between endpoints $x$ and $y$. This distribution is assumed to be Gaussian: $p(x_t|x, y) = \mathcal{N}\left(\alpha_t x + \beta_t y, \sigma_t^2 I\right)$, which allows $x_t$ to be directly sampled from a training pair $(x, y)$ using:

$$x_t = \alpha_t x_0 + \beta_t y + \sigma_t \epsilon, \tag{25}$$

where $\epsilon \sim \mathcal{N}(0, I)$; $\alpha_t$, $\beta_t$, $\sigma_t$ are time-dependent coefficients satisfying boundary conditions: $\alpha_0 = \beta_T = 1$ and $\alpha_T = \beta_0 = \sigma_0 = \sigma_T = 0$.

The transition distribution $p_\theta(x_{t-1}|x_t, y)$ for generating $x$ from $y$ (with $x_T \equiv y$) is modeled as $\mathcal{N}\left(\mu_{\theta, t, t-1}(x_t, y), \delta_{t-1|t}^2 \frac{\sigma_{t-1}^2}{\sigma_t^2} I\right)$ where the mean is given by (Kieu et al., 2025):

$$\mu_{\theta, t, t-1}(x_t, y) = \frac{\alpha_{t-1}}{\alpha_t} x_t + \left(\beta_{t-1} - \frac{\beta_t \alpha_{t-1}}{\alpha_t}\right) y + \left(\frac{\sigma_{t-1}\sqrt{\sigma_t^2 - \delta_{t-1|t}^2}}{\sigma_t} - \frac{\sigma_t \alpha_{t-1}}{\alpha_t}\right) \epsilon_\theta(x_t, t, y) \tag{26}$$

Here, $\delta_{t-1|t} \in [0, \sigma_t)$ controls the sampling variance and is typically defined as $\delta_{t-1|t} := \sqrt{\eta\left(\sigma_t^2 - \sigma_{t-1}^2 \frac{\alpha_{t-1}^2}{\alpha_t^2}\right)}$ with $\eta \in [0, 1]$ in the case of Brownian bridges (Li et al., 2023).

The noise prediction network $\epsilon_\theta$ in Eq. 26 is trained by minimizing the noise matching loss:

$$\mathcal{L}(\theta) = \mathbb{E}_{(x,y),t,\epsilon}\left[\mathbb{E}_{x_t}\left[\|\epsilon_\theta(x_t, t, y) - \epsilon\|_2^2\right]\right], \tag{27}$$

where $(x, y) \sim p(x, y)$, $t \sim \mathcal{U}(1, T)$, $\epsilon \sim \mathcal{N}(0, I)$.

Bidirectional diffusion bridges (Kieu et al., 2025) extend this framework to jointly model both $p(x|y)$ and $p(y|x)$ using a single shared noise prediction network.

# C ADDITIONAL RELATED WORK

Multi-domain translation (MDT) methods can be classified according to the underlying generative models. This section focuses on reviewing traditional VAE-based and GAN-based approaches.

## C.1 VAE-BASED METHODS

VAE-based methods generally aim to learn latent representations that facilitate translation across domains or modalities. JMVAE (Suzuki et al., 2016) captures shared representations with a joint encoder $q_\theta(z \mid x^1, x^2)$ and handles missing modalities at test time by aligning unimodal encoders $q_\theta(z \mid x^1)$ and $q_\theta(z \mid x^2)$ with the joint encoder through KL divergence minimization. TELBO (Vedantam et al., 2018) adopts a similar encoder design but differs in its training strategy. Instead of jointly optimizing all encoders, it first trains the joint encoder and then fits the unimodal encoders while keeping the joint encoder's parameters fixed. MFM (Tsai et al., 2019) factorizes the multimodal latent space into shared discriminative and modality-specific generative factors, enabling inference of missing modalities at test time from observed modalities. While these methods are sufficient for two modalities, they do not generalize to the truly multi-modal case.

More recently, MVAE (Wu & Goodman, 2018) innovatively factorize the joint encoder into a product of experts (PoE), i.e., $q_\Phi(z \mid x^1, x^2) = q_\phi(z \mid x^1) q_\phi(z \mid x^2) p(z)$, a notable advance that scales to multiple domains by training with randomly masked modalities and seamless inference when any modality is missing at test time. Alternately, MMVAE (Shi et al., 2019) constructs the joint encoder as a mixture of experts (MoE) of unimodal encoders, alleviating the precision-miscalibration issues inherent to PoE. Despite strong results in conventional MDT, both MVAE and MMVAE assume fully aligned tuples across domains, a requirement rarely satisfied in practice. In contrast, our method trains directly on domain pairs, removing the need for fully aligned triplets and improving practicality in real-world settings.

| Method | LPIPS ↓ | | | | | |
| --- | --- | --- | --- | --- | --- | --- |
| | Shoes-UMDT | | | Faces-UMDT-Latent | | |
| | Edge↔Shoe | Gray.↔Shoe | Edge↔Gray. | Ske.↔Face | Seg.↔Face | Ske.↔Seg. |
| StarGAN | 0.128/0.223 | 0.095/0.214 | 0.191/0.144 | - | - | - |
| Rectified Flow | 0.063/0.175 | 0.012/0.146 | 0.138/0.083 | 0.278/0.560 | 0.165/0.548 | 0.419/0.274 |
| Unidiffuser | 0.066/0.170 | 0.019/0.091 | 0.090/0.091 | 0.263/0.483 | 0.165/0.511 | 0.393/0.223 |
| iDR | **0.050**/0.129 | **0.003**/0.069 | **0.069/0.058** | **0.221/0.427** | 0.129/0.471 | **0.377/0.177** |
| dDR | **0.050**/0.128 | 0.004/**0.063** | 0.077/0.090 | 0.222/0.428 | **0.126/0.466** | 0.392/0.192 |

Table 4: LPIPS scores of our method and baselines on Shoes-UMDT and Faces-UMDT-Latent. Translations without paired data are marked in brown. The best results are shown in **bold**, and the second-best are underlined.

## C.2 GAN-BASED METHODS

Another line of research uses GAN (Goodfellow et al., 2014) for MDT. Pix2pix (Isola et al., 2017) uses paired datasets to train a conditional GAN with a reconstruction loss, aligning outputs with ground-truth targets while encouraging realism. CoGAN (Liu & Tuzel, 2016) instead learns a joint distribution across domains by sharing weights in high-level layers, enabling related samples across domains without aligned pairs. However, weight sharing constrains model design and limits scalability to high-resolution images. A popular approach for MDT is to leverage cycle-consistency constraint when training cross-domain translators (Zhu et al.; Kim et al., 2017; Yi et al., 2017; Liu et al., 2017), enforcing a meaningful relation between the input and the translated image. Nevertheless, these GAN-based methods do not scale gracefully with more domains, leading to higher computational cost.

Notably, StarGAN (Choi et al., 2018) introduces a unified GAN framework for MDT that uses a single generator and discriminator with an auxiliary domain classifier during training. Its successor, StarGAN-2 (Choi et al., 2020), injects continuous style codes via AdaIN (Huang & Belongie, 2017) to deliver diverse outputs. Like StarGAN, UFDN (Liu et al., 2018) also offers a unified MDT model and and extends the framework to disentangle domain-invariant features. More recently, MultimodalGAN (Zhu et al., 2024) proposes an MDT framework trained on fully aligned tuples across domains.

Our work is related to StarGAN in that both introduce frameworks for MDT, but differs in supervision and translation. We exploit paired datasets for training and perform indirect translations between domains lacking training pairs. In contrast, StarGAN treats such translations as an unpaired translations. We empirically show that leveraging paired supervision leads to more faithful input–output mappings and superior performance.

## D    ADDITIONAL EXPERIMENT

### D.1    ADDITIONAL IMPLEMENTATION DETAILS

#### D.1.1    DIFFUSION ROUTER

We adopt the standard Diffusion Router variant with DDPM (Ho et al., 2020) as the underlying diffusion process. A comparison with the bridge-based variant is provided in Appdx. D.3.5. The noise prediction network $\epsilon_\theta$ has a U-Net architecture following (Dhariwal & Nichol, 2021). For Shoes-UMDT (64×64) and Faces-UMDT-Pixel (128×128), $\epsilon_\theta$ operates directly on raw images with 128 and 256 base channels, respectively. For Faces-UMDT-Latent, COCO-UMDT-Star, and COCO-UMDT-Chain, the 256×256 images are first encoded into 32×32×4 latents using a pretrained VAE (Rombach et al., 2022), which are then processed by $\epsilon_\theta$ with 128 base channels. To ensure efficient training within a single H100-80GB GPU, we use a batch size of 128 for the 128-channel U-Net variant and 32 for the 256-channel variant. iDR is trained using AdamW (Loshchilov & Hutter, 2019) with a learning rate of 1e-4, $\beta_1 = \beta_2 = 0.9$ and 3000 warm-up steps across all datasets. The number of training steps is 250k for Shoes-UMDT and Faces-UMDT, and 500k for COCO-UMDT.

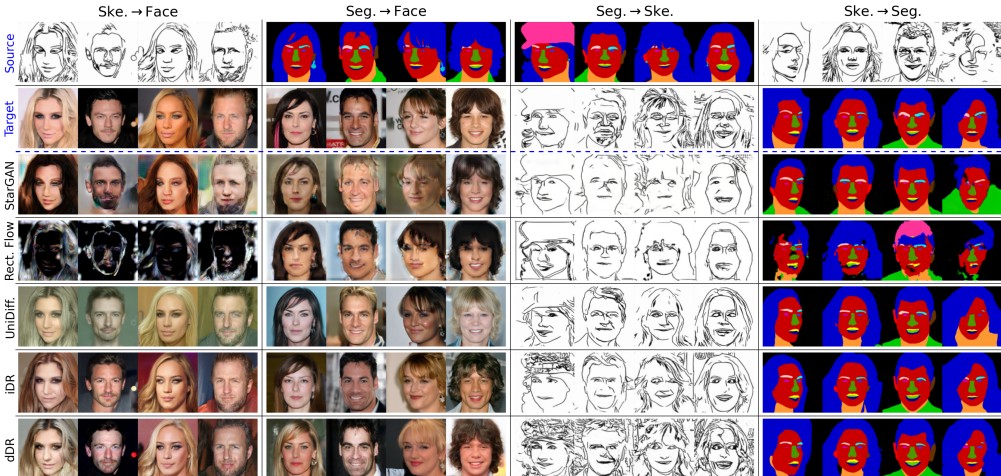

Figure 4: Qualitative results of our method and baselines on Faces-UMDT-Pixel.

Training takes roughly three to four days. dDR is finetuned from a pretrained iDR with a reduced learning rate of 5e-5, loss coefficients $\lambda_1 = 1$ and $\lambda_2 = 1$, and 5 refinement steps. The number of finetuning steps is 100k for Shoes-UMDT and Faces-UMDT, and 150k for COCO-UMDT. We run DDIM (Song et al., 2021a) with 1000 steps to translate from one domain to another.

### D.1.2 BASELINES

We implement StarGAN using its official repository. While more recent GAN-based approaches for MDT exist, such as MultimodalGAN (Zhu et al., 2024), we exclude them from our baselines due to the lack of publicly available code and instead discuss them in Related Work.

UniDiffuser (Bao et al., 2023) is designed to model noisy samples across all domains at random noise levels. Since our training data provide only domain pairs rather than full tuples, we substitute missing domains with Gaussian noise and set the corresponding domain-specific time step to 999 during training. Rectified Flow (Liu et al., 2022), which natively supports only two-domain translation with each domain as a boundary distribution, is adapted to the multi-domain setting by conditioning the velocity network on both source and target domain labels, similar to our approach. For fairness, we implement UniDiffuser and Recified Flow using the same architecture and settings as our method.

### D.2 ADDITIONAL RESULTS

### D.2.1 SHOES-UMDT AND FACES-UMDT-LATENT

Table 4 shows the LPIPS scores for our method and the baselines. Overall, iDR and dDR consistently outperform the baselines across all translation tasks. On tasks with paired supervision, both methods achieve comparable results. For translations without paired data, dDR exhibits a slight performance drop relative to iDR, but supports direct non-central translations. These observations align with the FID results presented in the main text.

### D.2.2 FACES-UMDT-PIXEL

For completeness, we compare iDR and dDR against baselines on Faces-UMDT-Pixel (see Table 5). The results from Table 5 mirror the trends in our main experiments at Section 5.2, with iDR consistently surpassing all baselines and showing the largest gains on high uncertainty translations (e.g., Ske.→Face and Seg.→Face). For Rectified Flow (RF), we observe poor generalization on the test set, particularly on Ske.→Face, yielding underperforming results. This underperformance likely arises because training uses FFHQ for Ske.→Face while testing use CelebA-HQ for Ske.→Face, causing a mismatch between the training and test distributions and encourging model's generalization, where RF fails.

| Method | FID↓ | | | LPIPS↓ | | |
|---|---|---|---|---|---|---|
| | Ske.↔Face | Seg.↔Face | Ske.↔Seg. | Ske.↔Face | Seg.↔Face | Ske.↔Seg. |
| StarGAN | 38.16/91.67 | 32.98/75.41 | 50.17/76.40 | 0.280/0.479 | 0.282/0.471 | 0.408/0.364 |
| Rectified Flow | 9.88/261.96 | 19.54/122.87 | 45.01/109.37 | 0.160/0.635 | 0.137/0.464 | 0.362/0.398 |
| Unidiffuser | 18.49/36.99 | 14.03/25.36 | 36.89/15.79 | 0.181/0.548 | 0.125/0.442 | 0.367/0.191 |
| iDR | 9.25/13.32 | 4.12/**10.54** | **13.51/3.88** | 0.159/**0.329** | **0.101/0.412** | **0.361/0.146** |
| dDR | **9.02/12.81** | **3.57**/12.88 | 28.46/3.91 | **0.139**/0.347 | 0.106/0.421 | 0.422/0.167 |

Table 5: Results on Faces-UMDT-Pixel of our method and baselines. Translations without paired data are marked in brown. The best results are shown in **bold**, and the second-best are underlined.

In comparison, dDR performs comparably to iDR on most translations while maintaining a clear margin over the baselines. We attribute dDR's lower results on Seg.→Ske. to FID's sensitivity for sketches: background differences and minor detail changes can depress the score even when outputs are visually similar to the targets (see Fig. 4). These results empirically indicate that iDR and dDR can generalize to high-resolution data.

## D.3 ABLATION STUDIES

Unless otherwise specified, ablation studies are conducted on Faces-UMDT-Latent to evaluate the impact of different design choices. To reduce both training and sampling costs, we substitute the large U-Net used in the main experiments (304.8M parameters) with a smaller variant (32.3M parameters).

| | $n$ | FID↓ | | |
|---|---|---|---|---|
| | | Ske.↔Face | Seg.↔Face | Ske.↔Seg. |
| iDR | - | 14.21/39.06 | 10.18/24.95 | 20.82/10.85 |
| dDR | 0 | 16.11/39.82 | 13.75/25.77 | 55.30/13.35 |
| | 1 | 14.14/40.61 | 11.77/23.65 | 37.45/13.13 |
| | 3 | 13.73/41.14 | 11.58/25.26 | 26.77/10.96 |
| | 5 | 13.52/37.26 | 11.52/22.73 | 26.27/11.37 |

| | $\lambda_2$ | FID↓ | | |
|---|---|---|---|---|
| | | Ske.↔Face | Seg.↔Face | Ske.↔Seg. |
| iDR | - | 14.21/39.06 | 10.18/24.95 | 20.82/10.85 |
| dDR | 0 | 405.12/306.61 | 355.78/307.15 | 403.19/328.75 |
| | 0.3 | 19.44/45.94 | 17.01/30.90 | 52.31/13.18 |
| | 1 | 16.11/39.82 | 13.75/25.77 | 55.30/13.35 |
| | 3 | 14.47/39.38 | 10.19/25.50 | 82.77/13.23 |

Table 6: FID scores of dDR finetuned on Faces-UMDT-Latent w.r.t. different number of refinement steps $n$.

Table 7: FID scores of dDR finetuned on Faces-UMDT-Latent w.r.t. different values of $\lambda_2$. Tweedie refinement is not applied in this case (i.e, $n = 0$).

### D.3.1 IMPACT OF THE NUMBER OF REFINEMENT STEPS $n$

We analyze the effect of the number of refinement steps $n$ n in the proposed Tweedie refinement (Eq. 12) by varying $n \in \{0, 1, 3, 5\}$, with results summarized in Table 6. When $n = 0$ (i.e,

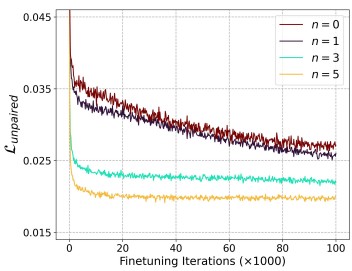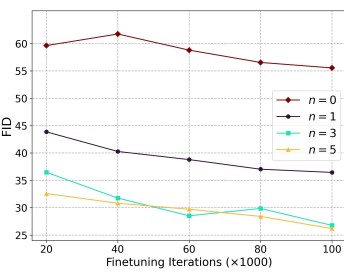

Figure 5: Learning curves of finetuned dDR on Face-UMDT-Latent w.r.t. different number of Tweedie refinement steps $n \in \{0, 1, 3, 5\}$. The task is Segment→Sketch translation.

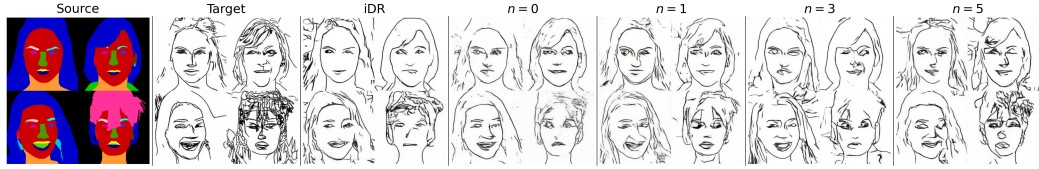

Figure 6: Comparison of translated images from Segment to Sketch w.r.t different value of $n$.

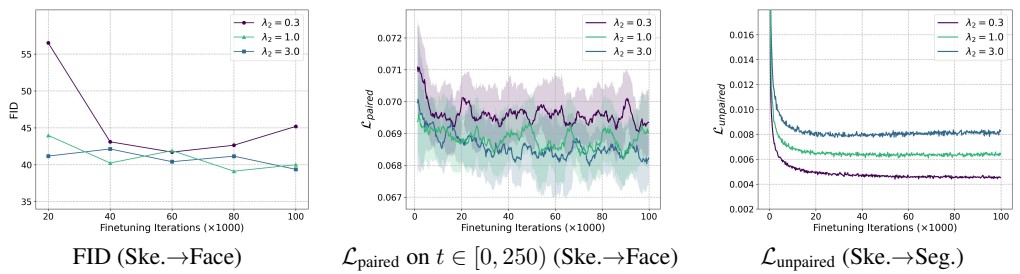

| FID (Ske.→Face) | $\mathcal{L}_{\text{paired}}$ on $t \in [0, 250]$ (Ske.→Face) | $\mathcal{L}_{\text{unpaired}}$ (Ske.→Seg.) |

Figure 7: Learning curves of dDR finetuned with different values of $\lambda_2$.

no refinement), dDR performs poorly on direct translations between non-central domains (e.g., Sketch↔Segment), though it remains comparable to iDR on translations involving central domains (e.g., Sketch↔Face, Segment↔Face). The failure is most pronounced for the Segment→Sketch translation task, where the target domain has limited detail and the FID scores are highly sensitive to minor variations. This indicates that without refinement, dDR struggles to capture the correct mapping from Segment to Sketch. Increasing $n$ consistently improves translation quality across all settings, particularly in unpaired translations, as reflected in the performance curves in Fig. 5 and the qualitative results in Fig. 6. The improvement arises because the refined noisy sample $x_{t,(n)}^j$ provides a closer approximation to samples from $p_{\text{ref}}\left(x_t^j | x^c\right)$, thereby yielding more accurate predictions from $\epsilon_{\text{ref}}\left(x_t^j, t, x^c, j, c\right)$ (Eq. 10). Consequently, the unpaired loss $\mathcal{L}_{\text{unpaired}}$ better approximates the variational bound in Eq. 9.

### D.3.2 IMPACT OF THE COEFFICIENT $\lambda_2$ IN THE LOSS $\mathcal{L}_{\text{FINAL}}$

We investigate the effect of the coefficient $\lambda_2$ by experimenting with different values in $\{0, 0.3, 1, 3\}$. As shown in Table 7, setting $\lambda_2 = 0$ causes the FID scores for all translation tasks to diverge. This happens because the unforgetting term $\mathcal{L}_{\text{paired}}$ is discarded from $\mathcal{L}_{\text{final}}$, causing the finetuned dDR to forget previously learned central↔non-central mappings (e.g., Sketch↔Face and Segment↔Face). Consequently, non-central↔non-central translations such as Sketch↔Segment are also learned incorrectly, since training them depends on noisy samples from the central↔non-central mappings. When $\lambda_2 > 0$, the fine-tuned model must balance preserving old translations with learning new ones. Increasing $\lambda_2$ improves the FID scores of the finetuned dDR on non-central↔non-central translations, enabling them to match those of the pretrained iDR, but at the cost of degrading performance on the central↔non-central translations. This trade-off is more clearly reflected in the learning curves in Fig. 7. Empirically, we found that $\lambda_2 = 1$ provides the best balance across translation tasks.

### D.3.3 IMPACT OF TRAINING FROM SCRATCH WITH $\mathcal{L}_{\text{FINAL}}$

We study the effect of training dDR from scratch versus fine-tuning it from a pretrained iDR, both strategies using roughly the same loss $\mathcal{L}_{\text{final}}$ in Eq. 11. In this experiment, we set $n = 3$. Training from scratch is computationally demanding–requiring over three days to run 200K iterations of a small U-Net on a single H100 GPU–so we limit the from-scratch run to 200K iterations and compare it with 100K fine-tuning iterations under identical settings. Fig. 8 presents the loss and FID curves for both strategies on central↔non-central and non-central↔non-central translation tasks. Overall, dDR trained from scratch performs poorly in the early stages (as indicated by high FID

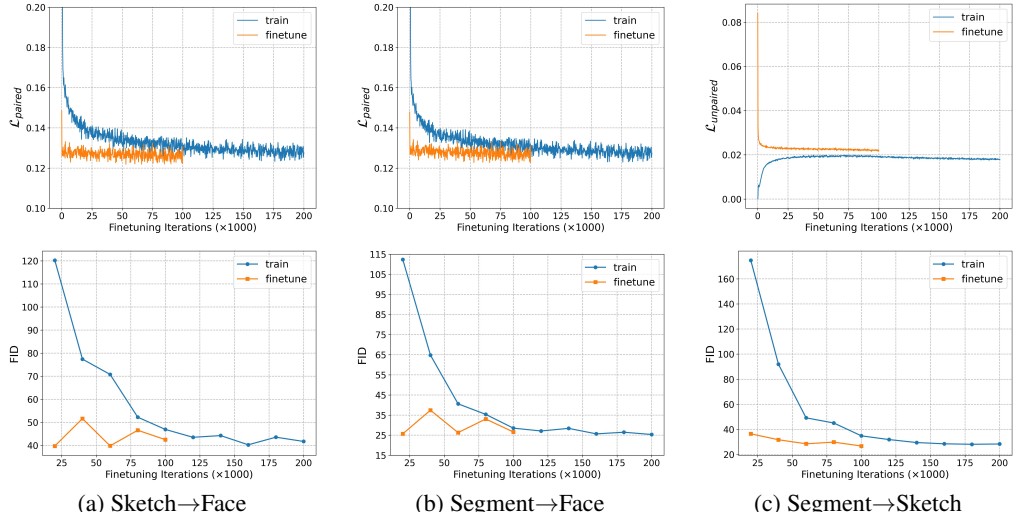

Figure 8: Loss and FID curves of trained-from-scratch and finetuned dDRs.

| NFE | **Faces-UMDT-Latent** | | **COCO-UMDT-Star** | | | | | |
| | Ske.↔Seg. | | Ske.↔Seg. | | Ske.↔Depth | | Seg.↔Depth | |
| | iDR | dDR | iDR | dDR | iDR | dDR | iDR | dDR |
| 10 | 74.33/7.95 | 39.29/6.65 | 98.14/79.17 | 31.29/34.19 | 78.65/13.96 | 27.10/12.09 | 79.95/24.97 | 35.31/16.48 |
| 20 | 42.25/6.98 | 23.79/5.82 | 50.09/32.07 | 28.26/25.67 | 42.95/11.30 | 22.69/11.26 | 32.93/15.35 | 26.03/15.63 |
| 30 | 31.77/6.67 | 22.31/5.72 | 36.01/25.99 | 27.93/24.81 | 31.37/9.83 | 22.09/10.35 | 26.12/14.71 | 25.57/15.02 |
| 50 | 23.37/6.39 | 21.85/5.61 | 27.84/23.45 | 27.19/24.06 | 19.39/9.03 | 21.65/9.59 | 24.87/13.93 | 25.21/14.74 |
| 100 | 18.05/6.25 | 20.82/5.58 | 24.70/23.16 | 26.87/23.78 | 18.81/8.79 | 21.18/9.29 | 24.08/12.49 | 25.18/14.62 |
| 1000 | 16.17/6.19 | 19.42/5.52 | 23.06/23.01 | 26.73/23.64 | 18.15/8.86 | 20.75/9.42 | 23.37/12.11 | 24.91/14.87 |

Table 8: FID across NFE for non-central↔non-central translations with iDR and dDR on Faces-UMDT-Latent and COCO-UMDT-Star.

scores), but gradually improves and eventually approaches the performance of the fine-tuned variant as training converges. Interestingly, the $\mathcal{L}_{\text{unpaired}}$ curve for from-scratch training behaves differently from $\mathcal{L}_{\text{paired}}$: it increases from near zero rather than decreasing from a large value. This occurs because, at the start of training, $\epsilon_\theta$ is untrained, making the expectations of $\epsilon_\theta \left( x_t^j, t, x^i, j, i \right)$ and of $\epsilon_\theta \left( x_t^j, t, x^c, j, c \right)$ approximately equal.

### D.3.4 COMPARISON OF iDR AND dDR ACROSS SAMPLING STEPS

We evaluate iDR and dDR on Faces-UMDT-Latent and COCO-UMDT-Star while varying the total number of sampling steps (NFE) from 10 to 1000. We focus on non-central↔non-central translations because the central↔non-central results for iDR and dDR are already comparable at matched NFE in the main experiments. Both methods use 304.8M-parameter U-Net models and are evaluated with the same total NFE for each non-central↔non-central translation.

From Table 8, we observe that iDR's performance deteriorates sharply as NFE decreases, whereas dDR remains much more stable, with only a moderate drop in FID. At NFE ≤ 30, dDR achieves 50–100% lower FID scores than iDR on several translation tasks, such as Seg.→Ske. on Faces-UMDT-Latent, and Ske.↔Seg., Ske.←Depth, and Seg.←Depth on COCO-UMDT-Star. This discrepancy arises because iDR relies on an intermediate central-domain sample whose quality degrades significantly when NFE is small, thereby impairing the final non-central-domain output. By contrast, dDR directly generates the non-central domain and thus avoids this issue. These results strongly indicate

| Model type | Model | FID↓ | | | LPIPS↓ | | |
|---|---|---|---|---|---|---|---|
| | | Ske.↔Face | Seg.↔Face | Ske.↔Seg. | Ske.↔Face | Seg.↔Face | Ske.↔Seg. |
| Bridge | BBDM | 25.69/41.79 | 19.02/27.76 | 27.92/20.76 | 0.244/0.457 | 0.144/0.492 | 0.399/0.196 |
| | I²SB | 16.56/31.81 | 10.58/24.56 | 21.37/13.16 | 0.241/0.456 | 0.153/0.492 | 0.396/0.208 |
| | DDBM | 15.51/42.55 | 10.32/27.77 | 19.51/11.41 | 0.246/0.477 | 0.156/0.506 | 0.408/0.213 |
| | BDBM | 13.67/33.98 | 6.12/26.73 | 24.88/6.63 | 0.244/0.479 | 0.151/0.500 | 0.382/0.204 |
| Diffusion | DDPM | **9.07/23.88** | **6.12/19.12** | 15.37/6.15 | **0.221/0.427** | **0.129/0.471** | 0.377/0.177 |

Table 9: Results on Faces-UMDT-Latent comparing diffusion-based and bridge-based DRs.

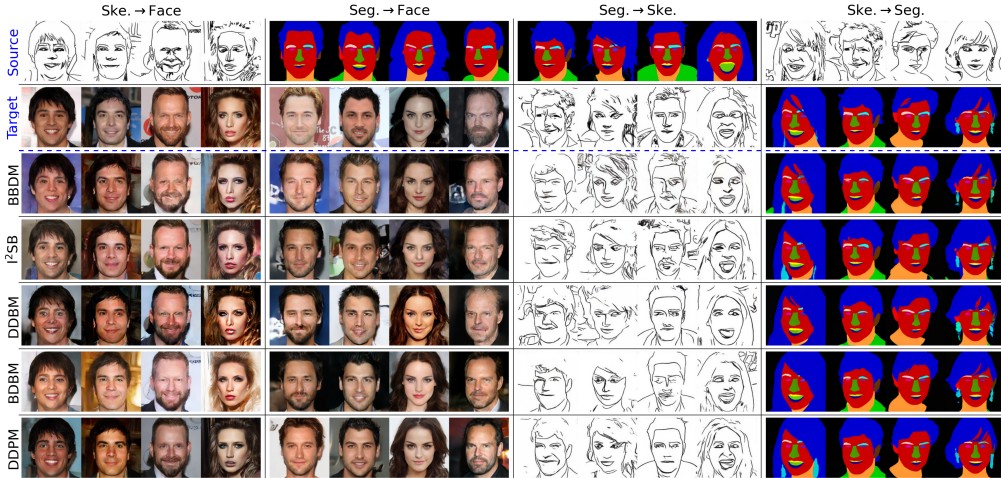

Figure 9: Qualitative comparison of diffusion-based and bridge-based DRs on Faces-UMDT-Latent.

that dDR is not sensitive to the central domain sample quality and is the clearly preferable choice when sampling with a limited number of steps.

### D.3.5 COMPARISON OF STANDARD AND BRIDGE-BASED DRS

We compare diffusion-based against bridge-based DRs on Faces-UMDT-Latent. For diffusion-based DR, we adopt DDPM schedules with the noise parameterization. For bridge-based DR, we follow schedules and parameterizations of state-of-the-art bridges, including BBDM (Li et al., 2023), I²SB (Liu et al., 2023a), DDBM (Zhou et al., 2024), and BDBM (Kieu et al., 2025). In all cases, a single 304.8M-parameter U-Net models all conditionals between the central and each non-central domain; translations between two non-central domains are executed indirectly via the central domain.

Table 9 shows that the diffusion-based DR outperforms bridge variants on most translation directions, with especially large margins on high-variance mappings (e.g., Ske.→Face, Seg.→Face). We attribute this advantage to diffusion's Gaussian prior and iterative denoising, which recover missing detail under sparse conditioning (e.g., sketches), whereas bridge-based sampling tightly anchors to the input and can suppresses plausible completions. Consistent with this, Fig. 9 shows that bridge-based DRs on Ske.→Face often over-condition on noisy sketch details, producing unrealistic artifacts. Moreover, diffusion models partially share generative processes across domains (from a common Gaussian), while bridge models learn separate source-to-target couplings for each paired dataset. A single U-Net thus faces a harder multi-coupling approximation under bridge designs, particularly when paired datasets differ substantially. Overall, these results indicate that diffusion-based DRs are better suited to MDT and can be fine-tuned to support direct non-central translations without paired supervision.

### D.3.6 IMPACT OF THE LEARNING RATE FOR FINETUNING

Table 10 examines the learning-rate trade-off when finetuning iDR into dDR for direct translation. Higher rates accelerate learning of the new Ske.↔Seg. mappings but significantly degrade the pre-

| | lr | FID↓ | | |
|---|---|---|---|---|
| | | Ske.↔Face | Seg.↔Face | Ske.↔Seg. |
| IDR | - | 14.21/39.06 | 10.18/24.95 | 20.82/10.85 |
| dDR | 1e-4 | 17.05/43.18 | 15.16/28.90 | 52.41/12.87 |
| | 5e-5 | 16.11/39.82 | 13.75/25.77 | 55.30/13.35 |
| | 1e-5 | 15.55/39.56 | 13.60/25.13 | 71.09/15.09 |
| | 5e-6 | 14.58/40.01 | 13.51/24.58 | 76.96/16.50 |
| | 1e-6 | 14.90/39.94 | 13.01/25.12 | 82.04/18.84 |

Table 10: FID on Faces-UMDT-Latent for dDR across finetuning learning rates. All dDR models were fine-tuned without Tweedie refinement ($n = 0$).

trained mappings, e.g., Ske.↔Face and Seg.↔Face, relative to iDR. At 1e-4, dDR achieves the best FID on Ske.↔Seg. yet yields the worst FIDs on Ske.↔Face and Seg.↔Face compared with iDR and dDR finetuned at lower rates. Reducing the rate reduces catastrophic forgetting and restores the pretrained directions, reaching 14.90/39.94 with Ske.↔Face and 13.01/25.12 on Seg.↔Face at 1e-6. This improvement comes at the cost of the direct translation mappings, with Ske.→Seg. FID increasing from 52.41 to 82.04 and Seg.→Ske. FID increasing from 12.87 to 18.84. Overall, the results show a clear trade-off between retaining pretrained tasks and learning the new one. Among the tested settings, 5e-5 offers the best balance.

## D.4 QUALITATIVE RESULTS ON COCO-UMDT-STAR AND COCO-UMDT-CHAIN

We visualize dDR's results on COCO-UMDT-Star and COCO-UMDT-Chain to illustrate its ability to handle UMDT. Specifically, Figs. 10, 11, 12, and 13 show cross-domain samples conditioned on color images, sketches, depth maps, and segmentation maps, respectively. Because segmentation maps are not ideally suited to a continuous diffusion process in latent space and predicted class labels are hard to decode perfectly, we apply DBSCAN (Ester et al., 1996) as a post-processing step that merges noisy pixels into the nearest region with the smallest label difference.

Input Color      Color → Ske.      Color → Seg.      Color → Depth

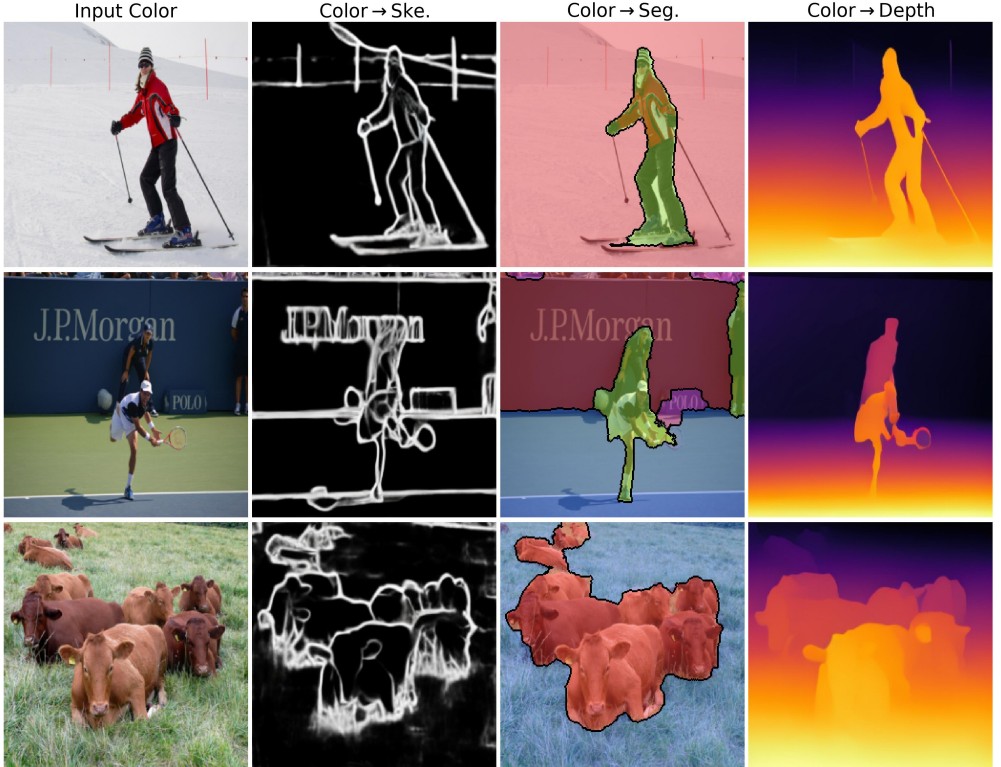

(a) Results on COCO-UMDT-Star. The classes for each segmentation map (from top to bottom) are: (snow, person), (wall, person, floor), (cow, grass).

Input Color      Color → Ske.      Color → Seg.      Color → Depth

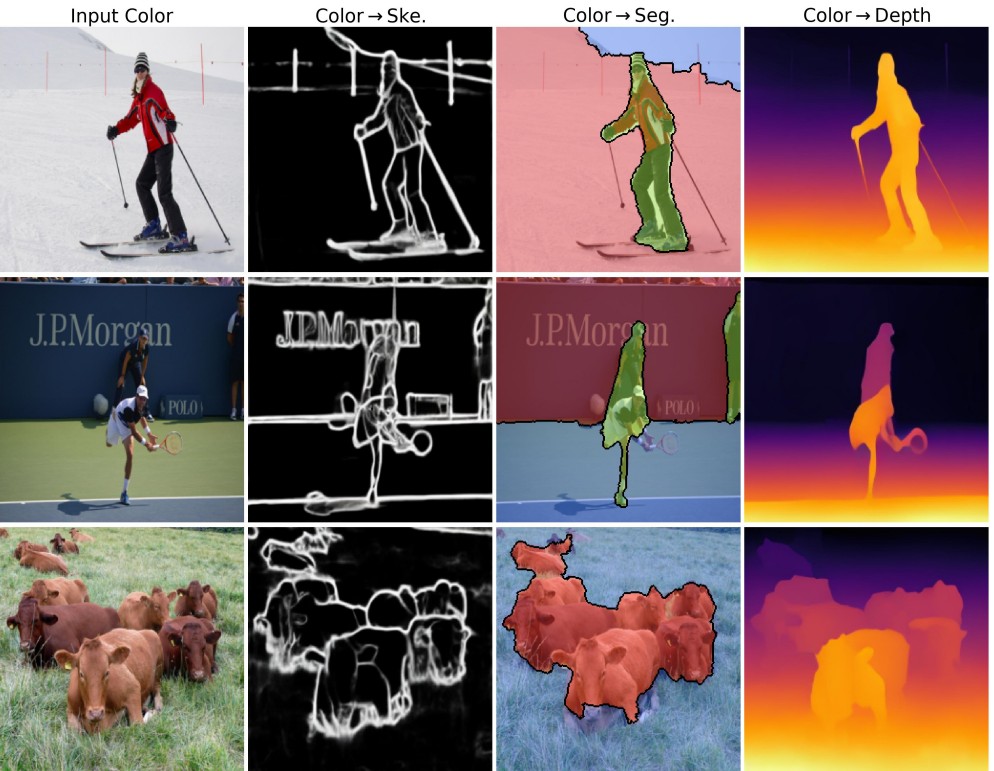

(b) Results on COCO-UMDT-Chain. The classes for each segmentation map (from top to bottom) are: (snow, person, sky), (wall, person, floor), (cow, grass).

Figure 10: Results of finetuned dDR on COCO-UMDT-Star (a) and COCO-UMDT-Chain (b) for translation tasks from Color domain to Sketch, Segment, and Depth domains.

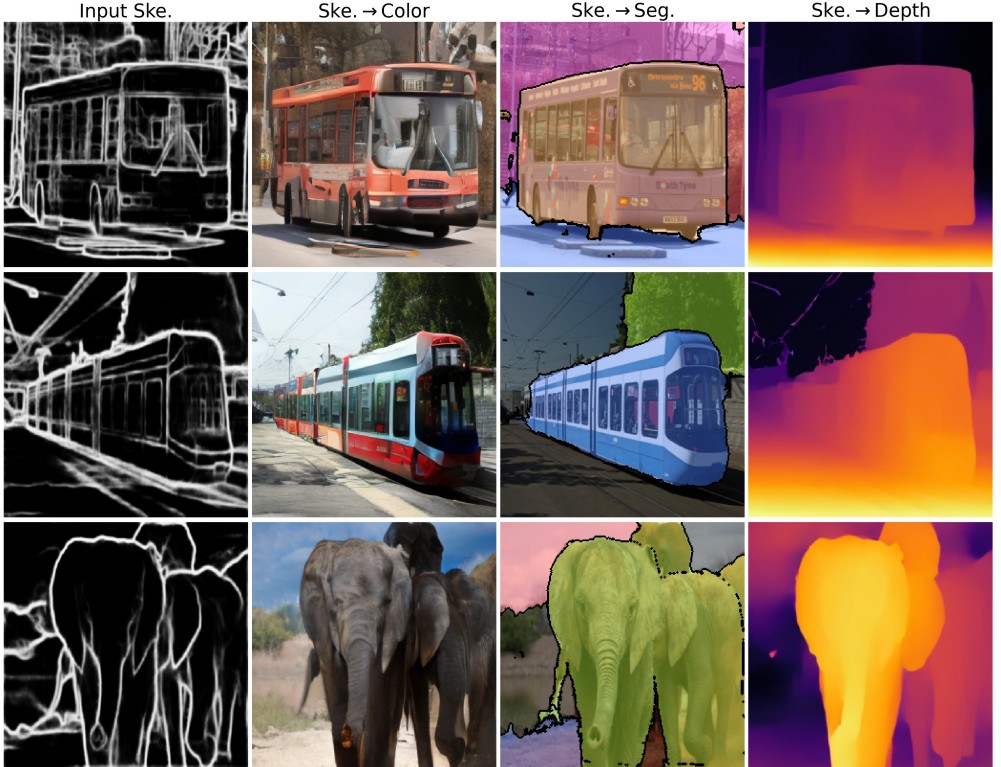

(a) Results on COCO-UMDT-Star. The classes for each segmentation map (from top to bottom) are: (building, bus, road, bush), (tree, train), (elephant, sky, gravel).

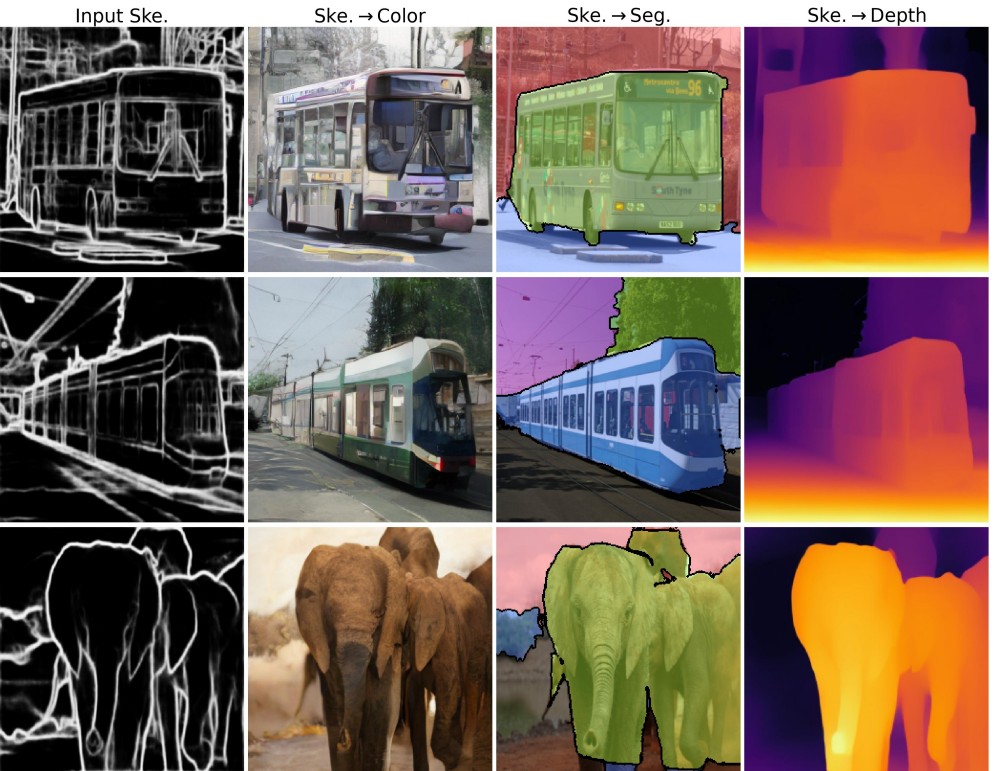

(b) Results on COCO-UMDT-Chain. The classes for each segmentation map (from top to bottom) are: (bus, road, wall), (tree, train, sky), (elephant, sky, tree).

Figure 11: Results of finetuned dDR on COCO-UMDT-Star (a) and COCO-UMDT-Chain (b) for translation tasks from Sketch domain to Color, Segment, and Depth domains.

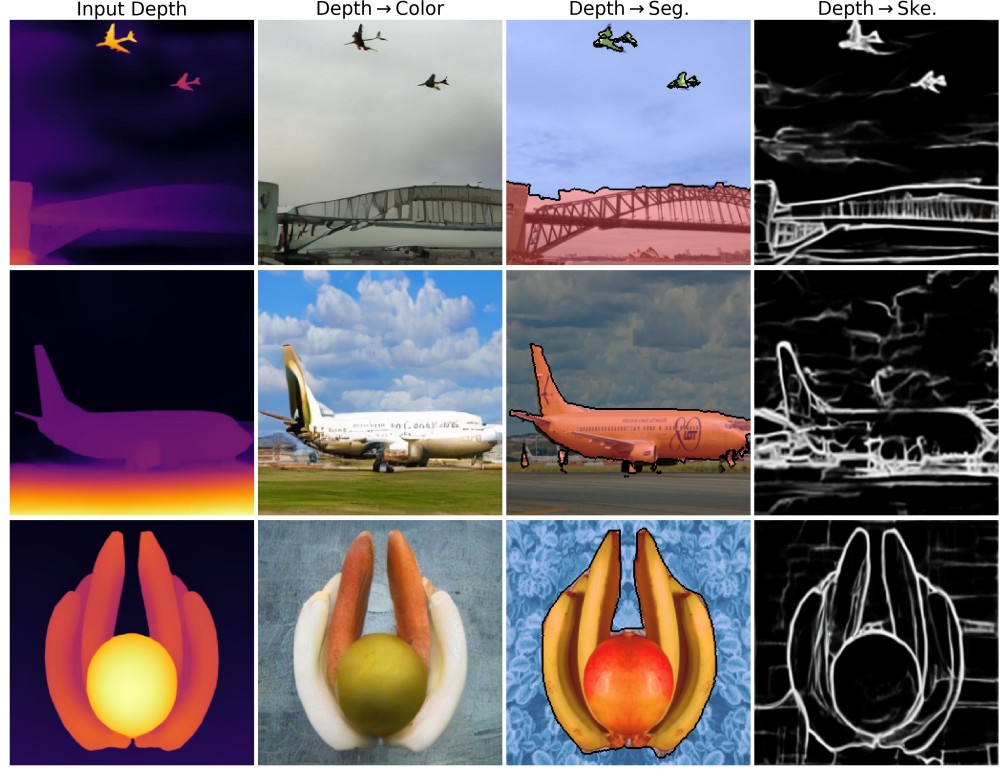

(a) Results on COCO-UMDT-Star. The classes for each segmentation map (from top to bottom) are: (sky, airplane, bridge), (airplane), (banana, table).

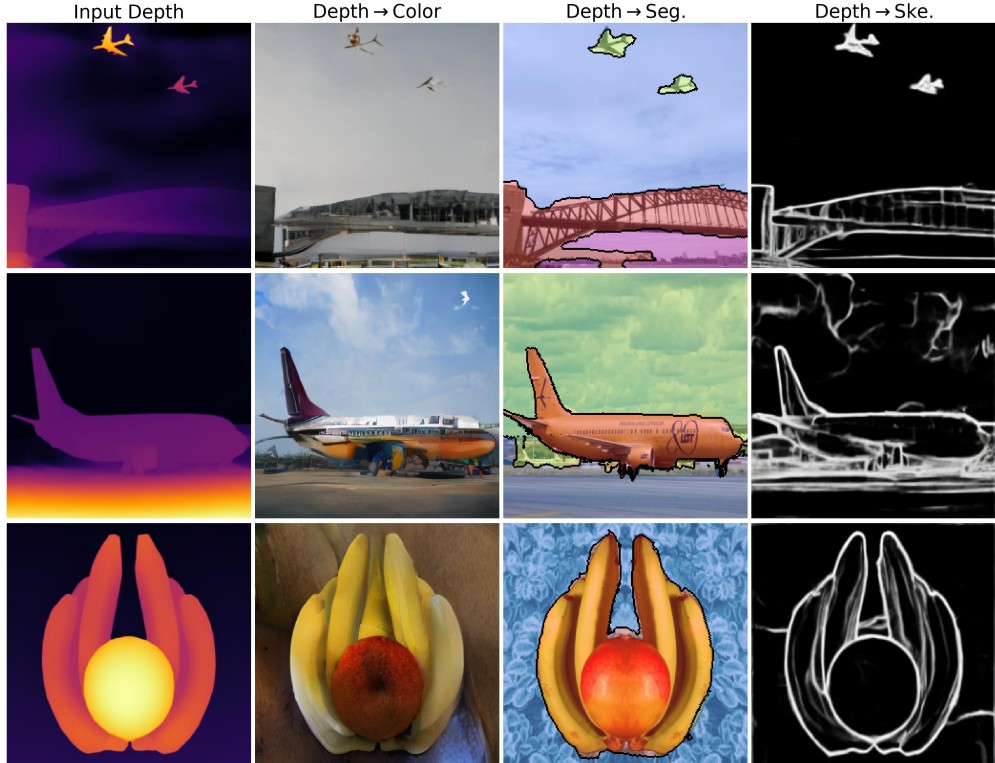

(b) Results on COCO-UMDT-Chain. The classes for each segmentation map (from top to bottom) are: (bridge, airplane, sky, undefined), (airplane, sky, road), (banana, floor).

Figure 12: Results of finetuned dDR on COCO-UMDT-Star (a) and COCO-UMDT-Chain (b) for translation tasks from Depth domain to Color, Segment, and Sketch domains.

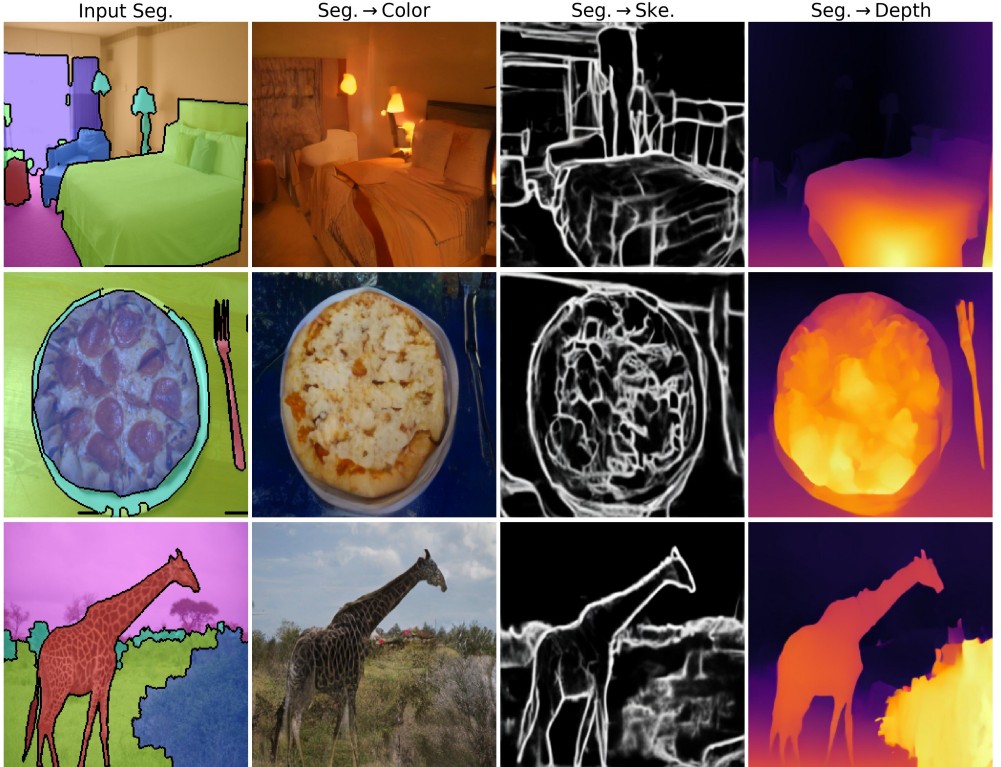

(a) Results on COCO-UMDT-Star. The classes for each segmentation map (from top to bottom) are: (bed, floor, window, light, wall), (pizza, undefined, table, fork), (grass, giraffe, bush, sky).

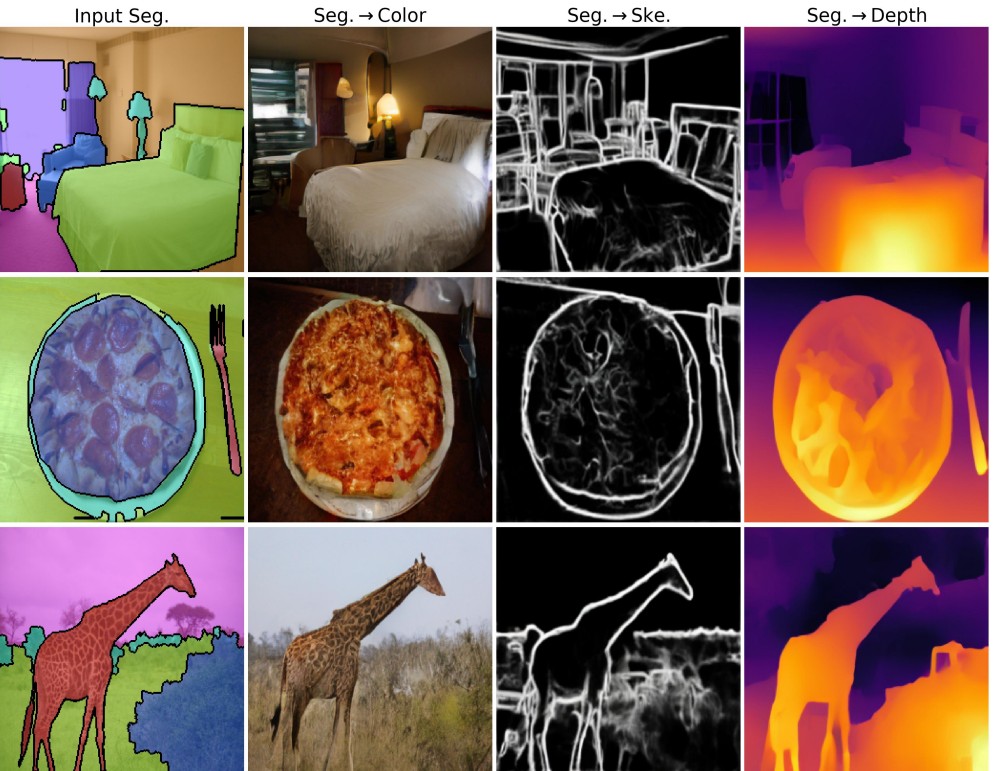

(b) Results on COCO-UMDT-Chain. The classes for each segmentation map (from top to bottom) are: (bed, floor, window, light, wall), (pizza, undefined, table, fork), (grass, giraffe, bush, sky).

Figure 13: Results of finetuned dDR on COCO-UMDT-Star (a) and COCO-UMDT-Chain (b) for translation tasks from Segment domain to Color, Sketch, and Depth domains.

