# OpenReview forum: "Universal Multi-Domain Translation via Diffusion Routers"
_ICLR.cc/2026/Conference — ICLR 2026 Poster_

### Official Review · Reviewer_es9u · 2025-10-29

**Soundness:** 3
**Presentation:** 3
**Contribution:** 3
**Rating:** 6
**Confidence:** 3

**Summary:**

This paper introduces universal Multi-Domain Translation, a problem setting for learning translation between any pair of K domains using only K-1 paired datasets. Specifically, this paper proposes a unified diffusion-based framework that avoids the need for training separate models for each domain pair. The framework is presented in two variants: iDR that routes translations through the central domain, and a dDR that distills iDR’s indirect path distributions into direct non-central mappings via a variational-bound objective and Tweedie refinement. To validate this approach, the authors construct three new image-based benchmark datasets and demonstrate that DR achieves state-of-the-art performance.

**Strengths:**

1. The problem formulation is well-motivated, and the background is clearly explained.
2. Extensive experiments validate the effectiveness of the proposed method on the constructed star- and chain-structured image-domain conditional generation tasks.
3. The writing is generally clear.

**Weaknesses:**

1. The universal claim is not supported by intra-modality experiments. The authors motivate the problem with compelling cross-modal examples like image/text/audio translation. However, all experiments are within the image-to-image translation domain, and the current results only prove the method's capability as a multi-style image translation system.
2. The paper's contribution appears less like the introduction of a new paradigm and more like a new training setup for an existing problem. When viewed as a solution for multi-style image translation under a specific data-sparsity assumption, its conceptual novelty is reduced.
3. iDR relies on multi‑step inference, and its outputs can accumulate approximation and routing errors. The proposed training of dDR distills from iDR’s generations, thereby inheriting these accumulated errors, which bounds dDR’s quality.  In addition, although DR is a single network whose parameter is roughly independent of the number of domains K, the number of non-central directed pairs to cover grows roughly as O(k^2). Together with Tweedie refinement for conditional sampling, this implies that total training time requirements scale up significantly with K.

**Questions:**

Minor clarifications for presentation. The X/Y format is not explicitly defined in tables. Please add a sentence to the table captions clarifying that the two numbers correspond to the forward and backward translation directions, respectively (e.g., "A->B / B->A").

---

> ### Author Response · Authors · 2025-11-23
> **Response to Reviewer es9u**
>
> We thank Reviewer es9u for the valuable comments. We would like to address your concerns and questions in detail below:
>
> >**\[Weakness 1\]** "The universal claim is not supported by intra-modality experiments. The authors motivate the problem with compelling cross-modal examples like image/text/audio translation. However, all experiments are within the image-to-image translation domain, and the current results only prove the method's capability as a multi-style image translation system."
>
> **\[Answer\]** We would like to note that our intention is not to claim universal cross-modal capability based solely on vision experiments. Rather, the paper introduces UMDT as a problem formulation (Sec. 3; Fig. 1), motivated by real‐world scenarios such as image$\leftrightarrow$text$\leftrightarrow$audio pipelines. The experiments focus on image domains because no UMDT benchmarks currently exist, and extending to cross-modal translation settings requires non-trivial effort in data processing, modality-specific design, and training, which we consider a separate line of contribution rather than part of this paper’s core scope. As clearly stated in Sec. 7, extending Diffusion Router (DR) to large-scale multimodal (image/text/audio) UMDT is an exciting future direction that we explicitly leave as planned future work.
>
>
> >**\[Weakness 2\]** "The paper's contribution appears less like the introduction of a new paradigm and more like a new training setup for an existing problem. When viewed as a solution for multi-style image translation under a specific data-sparsity assumption, its conceptual novelty is reduced."
>
> **\[Answer\]** To the best of our knowledge, the UMDT problem is novel and has not been studied in prior work. Reviewer es9u's characterization of UMDT as merely *"multi-style image translation under a specific data-sparsity assumption"* is not appropriate because (i) UMDT is conceptually distinct from conventional multi-domain translation, which typically involving central$\leftrightarrow$non-central translation, and (ii) UMDT is a general formulation that is not restricted to the image domain. Consequently, no existing method provides an adequate solution to this problem, which highlights both the novelty and the significance of our proposed DR. As demonstrated in our experiments, DR substantially outperforms all adapted multi-domain translation baselines under the UMDT setting.
>
> >**\[Weakness 3a\]** "iDR relies on multi-step inference, and its outputs can accumulate approximation and routing errors."
>
> **\[Answer\]** This is only true when we use a few number of denoising steps (e.g., \< 50) to translate from one domain to another. When the number of denoising steps is large (e.g., 1000), the accumulated errors in iDR is negligible, as evidenced by its strong performance across all benchmarks.
>
> >**\[Weakness 3b\]** "The proposed training of dDR distills from iDR's generations, thereby inheriting these accumulated errors, which bounds dDR's quality."
>
> **\[Answer\]** dDR does not use iDR's generated target domain samples during distillation. Instead, it relies on real samples from the training data. Therefore, dDR does not inherit accumulated errors from iDR's generated outputs. The primary factors influencing dDR's quality are the bias introduced when deriving its scalable training objective and the number of Tweedie refinement steps as discussed in Section 4.2, not any propagation of iDR's generation errors.
>
> >**\[Weakness 3c\]** "In addition, although DR is a single network whose parameter is roughly independent of the number of domains $K$, the number of non-central directed pairs to cover grows roughly as $O(K^{2})$. Together with Tweedie refinement for conditional sampling, this implies that total training time requirements scale up significantly with $K$."
>
> **\[Answer\]** The training time of dDR does not necessarily scale as $O(K^{2})$. During training, we only need to sample two random pairs $(x^{i}, x^{c}) \sim \mathcal D_{i,c}$ and $(x^{j},x^{c})\sim\mathcal{D}_{j,c}$ to learn the mapping between non-central domains $i$ and $j$, rather than enumerating all possible domain pairs. Tweedie refinement also uses a fixed number of steps, adding only a constant-factor overhead that does not grow with $K$. Consequently, the training time of dDR only grow only with the total dataset size$-$which is $O(K)$ under the assumption of $K-1$ paired datasets$-$like other methods.
>
> > Minor clarifications for presentation. The X/Y format is not explicitly defined in tables. Please add a sentence to the table captions clarifying that the two numbers correspond to the forward and backward translation directions, respectively (e.g., A$\to$B / B$\to$A).
>
> **\[Answer\]** We thank reviewer es9u valuable feedback. We will explicitly add a sentence to table caption to improve clarity.

---

> > ### Comment · Reviewer_es9u · 2025-11-26
> >
> > Thank you for your additional explanations. These responses have basically addressed my concerns. I will maintain the initial score and support the acceptance of this paper.

---

> > > ### Author Response · Authors · 2025-11-26
> > >
> > > We sincerely thank Reviewer es9u for the positive feedback and thoughtful comments. We are glad that our responses addressed your concerns and truly appreciate your support for acceptance.

---

### Official Review · Reviewer_ARK8 · 2025-10-30

**Soundness:** 4
**Presentation:** 3
**Contribution:** 3
**Rating:** 6
**Confidence:** 3

**Summary:**

This paper proposes a universal multi-domain translation model wherein a single model can be used to do pairwise translations across all the K considered domains with only requiring K-1 paired datasets.

**Strengths:**

- The proposed diffusion routers is novel
- The results are promising and outperforms baselines based on both qualitative and quantitative results
- They construct 3 different datasets and conduct extensive experiments on them to validate their approach

**Weaknesses:**

- The domains considered for all the datasets they constructed are quite limited. They only consider the image, segmentation, sketches, and depth, which are quite similar in structure only different appearance / details.
- The baselines compared are also quite old (2018, 2022, 2023). Granted that there are relatively few works on multi-domain translation models, I think it would still be good to also compare with more recent single domain translation models see how good its performance is compared to a dedicated model. This would also show whether or not training on more domains could improve performance due to having more auxiliary information.
- For the domains involved, the recent text-to-image models can also already achieve this kind of translation. It would be good to also compare with them since in a way it is also able to perform multi-domain translation with a single model.
- The authors motivated a lot of their work with image-text-audio domains. However, none of the experiments show any translations between these domains.

**Questions:**

- How does the performance of the model change depending on what you consider as the central domain? Currently, the central domain is always fixed.
- How well does the model perform on translating different modalities (audio-text-image) given that this is one of the motivating example in the introduction?
- Can the model work if the different domains contain images of different classes? For example consider translating between sketch-image-grayscale. For domain pair is image to sketch you have shoes as examples and for the image-grayscale you have faces. The central domain image contains both face and shoes. This setup satisfies your K-1 paired examples with overlap in the central domain. How well can the model now translate between the other domains?
- How about domains where structure is not preserved such as image to cartoon style? Current text-to-image models can generate a bunch of these paired examples, and different artist styles could also be considered as different domains.

---

> ### Author Response · Authors · 2025-11-23
> **Response to Reviewer ARK8 (1)**
>
> We thank Reviewer ARK8 for the insightful comments. We would like to address your concerns and questions in detail below:
>
> >**\[Weakness 1\]** "The domains considered for all the datasets they constructed are quite limited. They only consider the image, segmentation, sketches, and depth, which are quite similar in structure only different appearance / details."
>
> **\[Answer\]** Color images, segmentations, sketches, and depth maps are standard and widely used domains in paired and multi-domain translation research, as demonstrated in numerous prior works \[1, 2, 3, 4\]. Although these domains may have some similarities in structure, they nonetheless provide strong and meaningful benchmarks for evaluating UMDT. Our results clearly show that state-of-the-art multi-domain translation methods such as StarGAN, Rectified Flow, and UniDiffuser still fall significantly short of the performance achieved by our proposed Diffusion Router (DR) on these benchmarks.
>
> This indicates that the core difficulty of UMDT does not stem from structural differences between domains$-$which modern generative models, including diffusion models, can handle effectively$-$but rather *from the problem's formulation*: translating between two non-central domains without any paired training data. This setting is highly practical and is precisely what makes the UMDT problem both challenging and significant.
>
> \[1\] Image-to-Image Translation with Conditional Adversarial Networks, Isola et al., CVPR 2017
>
> \[2\] Multimodal Conditional Image Synthesis with Product-of-Experts GANs, Huang et al., ECCV 2022
>
> \[3\] Consistent Multimodal Generation via A Unified GAN Framework, Zhu et al., WACV 2024
>
> \[4\] One Diffusion to Generate Them All, Le et al., CVPR 2025
>
> >**\[Weakness 2a\]** "The baselines compared are also quite old (2018, 2022, 2023)."
>
> **\[Answer\]** To the best of our knowledge, the baselines we include are state-of-the-art methods within their respective categories for multi-domain translation, and we adapt them appropriately to the UMDT setting for comparison with our proposed methods. We would be glad to incorporate additional baselines if the reviewer can recommend suitable methods with publicly available implementations.
>
> >**\[Weakness 2b\]** "Granted that there are relatively few works on multi-domain translation models, I think it would still be good to also compare with more recent single domain translation models see how good its performance is compared to a dedicated model. This would also show whether or not training on more domains could improve performance due to having more auxiliary information."
>
> **\[Answer\]** For central$\leftrightarrow$non-central translations, DR is expected to perform comparable to or even better than dedicated paired-domain translation models because it leverages powerful conditional diffusion models to explicitly learn the corresponding conditional distributions involving the two domains. This conclusion follows directly from our theoretical analysis in Section 4.1, making additional experiments unnecessary. For non-central$\leftrightarrow$non-central translations, however, paired data do not exist by definition of the UMDT setting, meaning that conventional paired single-domain translators cannot be trained for these mappings at all. This is precisely the scenario where DR is needed and where it shows clear advantages over paired translation and even multi-domain translation baselines.
>
> While Reviewer ARK8's suggestion to examine whether training on additional domains could improve paired translation performance is interesting, it falls outside the scope of this paper.
>
> >**\[Weakness 3\]** "For the domains involved, the recent text-to-image models can also already achieve this kind of translation. It would be good to also compare with them since in a way it is also able to perform multi-domain translation with a single model."
>
> **\[Answer\]** Recent T2I diffusion models with visual controls (e.g., ControlNet \[5\], T2I-Adapter \[6\]) are mostly designed for non-central$\to$central generation in our terminology (sketch/segmentation/depth$\to$color images), not for non-central$\leftrightarrow$non-central translations (e.g., sketch$\leftrightarrow$segmentation) required by UMDT. In contrast, our work explicitly targets universal translation across all domain pairs, including these non-central$\leftrightarrow$non-central mappings.
>
> [5] Zhang et at., Adding Conditional Control to Text-to-Image Diffusion Models, ICCV 2023
>
> [6] Mou et al., T2I-Adapter: Learning Adapters to Dig out More Controllable Ability for Text-to-Image Diffusion Models, AAAI 2024

---

> ### Author Response · Authors · 2025-11-23
> **Response to Reviewer ARK8 (2)**
>
> >**\[Weakness 4\]**"The authors motivated a lot of their work with image-text-audio domains. However, none of the experiments show any translations between these domains."
>
> **\[Answer\]** Although UMDT is motivated by real-world multimodal translation across image-text-audio domains, the primary goals of this paper are to (i) formally define the UMDT problem, (ii) introduce DR as a scalable solution to it, and (iii) evaluate DR against strong multi-domain translation baselines while thoroughly analyzing its configurations on well-designed image-based UMDT benchmarks. As explained in our common response, extending DR to handle image-text-audio translation requires substantial additional work in data processing, model design, and training. We consider this a separate line of contribution, rather than part of the core contributions of the current paper.
>
> >**\[Question 1\]** "How does the performance of the model change depending on what you consider as the central domain? Currently, the central domain is always fixed."
>
> **\[Answer\]** The performance of translations between non-central domains may depends on the choice of central domain. Our assumption in Eq. 4 of Section 4.1 states that non-central domains are independent given the central one, which implies that non-central$\leftrightarrow$non-central translations improve when the central domain captures more information about the source non-central domain.
>
> >**\[Question 2\]** "How well does the model perform on translating different modalities (audio-text-image) given that this is one of the motivating example in the introduction?"
>
> **\[Answer\]**  As noted in our general response, this work focuses on the image setting to establish the UMDT problem and the DR framework. Extending UMDT and DR to the image-text-audio scenario is indeed feasible but would require substantial additional effort, warranting a dedicated follow-up study. We therefore leave this extension to future work.
>
> >**\[Question 3\]** "Can the model work if the different domains contain images of different classes? For example consider translating between sketch-image-grayscale. For domain pair is image to sketch you have shoes as examples and for the image-grayscale you have faces. The central domain image contains both face and shoes. This setup satisfies your $K-1$ paired examples with overlap in the central domain. How well can the model now translate between the other domains?"
>
> **\[Answer\]** Our UMDT formulation relies on semantic overlap via the central domain to ensure meaningful translations (lines 132$-$133). Paired datasets connected through the central domain are not required to share the same instances, but they *must share underlying semantic content*.
>
> In the setup you propose, the central "image" domain combines unrelated and non-overlapping classes, which are "shoes" and "faces". This violates the semantic overlap assumption, causing translations between the two non-central domains "sketch" and "grayscale" to be unattainable.
>
> >**\[Question 4\]** "How about domains where structure is not preserved such as image to cartoon style? Current text-to-image models can generate a bunch of these paired examples, and different artist styles could also be considered as different domains."
>
> **\[Answer\]** Our proposed DR has a domain-agnostic design and can be easily applied to the scenario suggested by Reviewer ARK8 where different styles are treated as different domains.

---

### Official Review · Reviewer_4zEu · 2025-10-31

**Soundness:** 3
**Presentation:** 3
**Contribution:** 3
**Rating:** 6
**Confidence:** 3

**Summary:**

They propose a new approach for multi-domain translation using a diffusion model. Unlike existing settings, where we assume the presence of many fully aligned tuples or unlabeled paired domains, they assume access to one central domain that has the paired samples with other domains. Then, they propose an approach that can achieve domain translation between unpaired domains by using the central domain as the bridge between the unpaired ones. Specifically, they propose an architecture that can translate between different image domains and training objectives. They conduct experiments on image translation and show that their approach outperforms existing diffusion based approaches with a large margin.

**Strengths:**

1. Their idea of using the central domain as the bridge between two unpaired domains sounds reasonable and novel. Their derivation of the objectives also looks reasonable.

2. The proposed architecture to achieve a single translation model, which specifies the origin and the target domain, also looks reasonable and might be novel.

3. This paper is easy to follow. Presentation is overall clear.

**Weaknesses:**

1. They repeat the discussion on translating image, text, and audio. However, they lack experiments on such cases. I wonder why they do not include such results. Also, my concern is that in such a modality translation setting, the proposed framework might not work well. They propose to apply a single network to exchange the domains of inputs. Since their experiments are conducted only in the translation of images, the effectiveness of swapping modality is not clear.

2. Lacking comparison to non-diffusion techniques in experiments. Readers should be curious about the performance difference between the proposed one and non-diffusion methods.

3. The novelty in the proposed method is not very clear to me. I guess the field of domain translation using a diffusion model is a popular topic. Since I am not familiar with this field, I could not figure out how novel the proposed method is, compared to the existing ones. They need a more careful discussion in explaining their approach in the introduction and method sections.

4. What if they have a designated model for one pair of domain translation and combine two models to bridge the domains? A single model still outperforms such an approach in terms of FID?

**Questions:**

Please answer the concerns described in the weaknesses.

---

> ### Author Response · Authors · 2025-11-23
> **Response to Reviewer 4zEu (1)**
>
> We thank Reviewer 4zEu for the thoughtful comments. We would like to address your concerns in detail below:
>
> >**\[Weakness 1a\]** "They repeat the discussion on translating image, text, and audio. However, they lack experiments on such cases. I wonder why they do not include such results."
>
> **\[Answer\]** As we clarified in our common response to all reviewers, applying the Diffusion Router framework to multimodal translation across image, text, and audio would require substantial computational resources, extensive data processing, careful model design, and large-scale training$-$an effort significant enough to constitute a standalone paper. Furthermore, our current contributions including (i) formulating the Universal Multi-Domain Translation (UMDT) problem, (ii) constructing the three large-scale visual UMDT datasets, (iii) proposing the Diffusion Router (DR) framework, and (iv) introducing a scalable training method that enables direct translation between non-central domains are already novel and substantial. Given constraints on time, computational resources, and to avoid diluting the focus of our core contributions, we chose not to include image-text-audio experiments in this paper and leave them for future work.
>
> >**\[Weakness 1b\]** "Also, my concern is that in such a modality translation setting, the proposed framework might not work well. They propose to apply a single network to exchange the domains of inputs. Since their experiments are conducted only in the translation of images, the effectiveness of swapping modality is not clear."
>
> **\[Answer\]** We would like to clarify that this concern of Reviewer 4zEu is unfounded. By representing samples from different modalities as *sequences of tokens in latent spaces*, it is technically possible to create DR with a single *transformer-based* network to translate between modalities. In fact, we had already developed preliminary ideas for extending the DR to image-text-audio translation during this work, but we chose to reserve these extensions for future research.
>
> >**\[Weakness 2\]** "Lacking comparison to non-diffusion techniques in experiments. Readers should be curious about the performance difference between the proposed one and non-diffusion methods."
>
> **\[Answer\]** We would like to clarify that our experiments already include comparisons against non-diffusion-based baselines, specifically StarGAN [1] (GAN-based) and Rectified Flow [2] (flow-based). As explained in Appendix D.1.2, our choice of baselines is constrained by (i) their compatibility with the UMDT problem and (ii) the availability of publicly accessible, reproducible code. Given these constraints, StarGAN and Rectified Flow are the strong non-diffusion-based baselines available for a fair and reliable comparison.
>
> [1] Choi et al., “StarGAN: Unified Generative Adversarial Networks for Multi-Domain Image-to-Image Translation”, CVPR 2018
>
> [2] Liu et al., “Flow Straight and Fast: Learning to Generate and Transfer Data with Rectified Flow”, ICLR 2023

---

> ### Author Response · Authors · 2025-11-23
> **Response to Reviewer 4zEu (2)**
>
> >**\[Weakness 3\]** "The novelty in the proposed method is not very clear to me. I guess the field of domain translation using a diffusion model is a popular topic. Since I am not familiar with this field, I could not figure out how novel the proposed method is, compared to the existing ones. They need a more careful discussion in explaining their approach in the introduction and method sections."
>
> **\[Answer\]** The novelty and significance of our work are clearly presented in the Introduction and summarized in lines 065$-$073. For convenience, we restate them here:
>
> +  **We formalize Universal Multi-Domain Translation (UMDT)**$-$a new and general setting that aims to learn translations between *any* pairs of $K$ domains using only $K-1$ paired datasets with a central domain.
>
> +  **We propose the Diffusion Router (DR)**, a unified diffusion-based framework that models all central$\leftrightarrow$non-central mappings using a *single* noise predictor.
>
> +  **We develop a scalable learning strategy** based on a variational-bound objective and Tweedie refinement, enabling direct non-central$\leftrightarrow$non-central translations without paired data.
>
> +  **We construct three new UMDT benchmarks** and show that DR achieves state-of-the-art performance for both indirect and direct translations.
>
> These contributions were recognized by other reviewers$-$for example, Reviewer AFk3 noted that *"the formalization of UMDT is a significant contribution"* and that "*the theoretical contribution for enabling dDR is clever,"* while Reviewer ARK8 wrote that *"The proposed diffusion routers is novel."*
>
> To the best of our knowledge, UMDT itself is a novel problem and has not been explored in prior work. It is highly relevant to practical scenarios$-$such as image$\leftrightarrow$text$\leftrightarrow$audio translation and multilingual translation$-$where only paired datasets with a central domain exist (as discussed in lines 044$-$046). Given the novelty of UMDT, the DR framework designed to address it is also novel. Its domain-agnostic design and scalable training strategy make it broadly applicable across UMDT settings. Furthermore, DR substantially outperforms state-of-the-art multi-domain translation baselines adapted to UMDT, as demonstrated in our experiments.
>
> >**\[Weakness 4\]** "What if they have a designated model for one pair of domain translation and combine two models to bridge the domains? A single model still outperforms such an approach in terms of FID?"
>
> **\[Answer\]** As discussed in Section 4.1 (lines 167-169), using separate models for every central$\leftrightarrow$non-central domain pair$-$as suggested by Reviewer 4zEu$-$would require up to $2(K-1)$ models, which is prohibitively expensive in both training time and storage. Our proposed DR addresses this issue by using a single model to handle all central$\leftrightarrow$non-central mappings, making the approach scalable to a large number of domains. In addition to being highly practical and efficient, DR is also theoretically well-motivated, as it explicitly models the couplings between central and non-central domains (i.e., $p\left(x^{k}|x^{c}\right)$ and $p\left(x^{c}|x^{k}\right)$). Therefore, it is entirely plausible that DR can outperform methods designed specifically for paired translation in terms of FID.

---

### Official Review · Reviewer_AFk3 · 2025-11-01

**Soundness:** 3
**Presentation:** 2
**Contribution:** 3
**Rating:** 4
**Confidence:** 3

**Summary:**

This paper introduces Universal Multi-Domain Translation (UMDT), a novel and practical problem setting for translating between any pair of K domains using only K-1 paired datasets structured around a central domain. To address this, the authors propose Diffusion Router (DR), a unified diffusion model that conditions a single noise predictor on both source and target domain labels to handle all translations involving the central domain. The authors create new large-scale benchmarks to evaluate their method, demonstrating that iDR outperforms existing baselines and that dDR can achieve direct translation, albeit with a trade-off in performance versus computational efficiency.

**Strengths:**

- The formalization of UMDT is a significant contribution. It addresses a critical limitation of existing multi-domain translation methods, which either require impractical fully-aligned data or are limited to a hub-and-spoke translation model. This problem setting is highly practical and forward-looking, with clear real-world applications.
- The core idea of the Diffusion Router is an elegant and parameter-efficient solution for the standard central-to-non-central translation task. The strong performance of iDR against other baselines validates this architectural choice and establishes it as a powerful model in its own right.
- The theoretical contribution for enabling dDR is clever. The formulation of a variational upper bound (Eq. 9) to align the direct and indirect translation paths is a non-trivial approach to learning from "pseudo-supervision" generated by the model's own indirect capabilities.

**Weaknesses:**

- The central weakness of this paper is that the main technical contribution, dDR, consistently performs worse than the simpler, two-step iDR baseline on the very tasks it was designed to improve. This is evident across all three benchmarks, where iDR's FID scores are superior to dDR's.The paper motivates dDR by claiming it overcomes the computational expense and potential quality degradation of the two-step iDR process. However, the results show that dDR introduces quality degradation. This undermines its primary justification and calls into question whether the proposed learning strategy is genuinely effective.
- While dDR is computationally cheaper at inference, the paper does not adequately quantify this benefit against the measured drop in quality. The faster inference speed is valuable, but if it comes at the cost of a significant increase in FID score, iDR remains the superior method in terms of output quality. The paper would be much stronger if it provided a clear analysis of this trade-off, perhaps identifying scenarios where the speed of dDR is worth the quality sacrifice.
- The authors acknowledge that their tractable objective relies on a single-sample Monte Carlo estimate inside a logarithm, which introduces bias. This technical compromise is a likely source of dDR's underperformance. The work would be more convincing if it included analysis isolating the effect of this bias.

**Questions:**

See Weakness

---

> ### Author Response · Authors · 2025-11-23
> **Response to Reviewer AFk3 (1)**
>
> We thank Reviewer AFk3 for the positive views regarding the strengths of our paper. We would like to address the weaknesses in detail below:
>
> >**[Weakness 1a]** “The central weakness of this paper is that the main technical contribution, dDR, consistently performs worse than the simpler, two-step iDR baseline on the very tasks it was designed to improve. This is evident across all three benchmarks, where iDR's FID scores are superior to dDR's.”
>
> **\[Answer\]** We respectfully disagree with Reviewer AFk3's characterization of iDR as a *"baseline"* and dDR as *"the main technical contribution"*. *Both iDR and dDR are our proposed methods* within the unified Diffusion Router framework, and are designed to be *complementary*. iDR is a *quality-first* approach that performs indirect translation through the central domain, whereas dDR is an *efficiency-oriented* variant that enables direct sampling between non-central domains to reduce computational cost. The observation that dDR performs slightly worse than iDR is *fully expected*. Indeed, it empirically confirms our theoretical analysis in Section 4.2 and therefore should not be interpreted as a weakness of our paper.Importantly, both iDR and dDR substantially outperform existing baselines including StarGAN [1], Rectified Flow [2] and UniDiffuser [3], highlighting the significance and effectiveness of our proposed methods.
>
> [1] Choi et al., “StarGAN: Unified Generative Adversarial Networks for Multi-Domain Image-to-Image Translation”, CVPR 2018
>
> [2] Liu et al., “Flow Straight and Fast: Learning to Generate and Transfer Data with Rectified Flow”, ICLR 2023
>
> [3] Bao et al., “One Transformer Fits All Distributions in Multi-Modal Diffusion at Scale” ICML 2023
>
> >**[Weakness 1b]** “The paper motivates dDR by claiming it overcomes the computational expense and potential quality degradation of the two-step iDR process. However, the results show that dDR introduces quality degradation. This undermines its primary justification and calls into question whether the proposed learning strategy is genuinely effective.”
>
> **\[Answer\]** In our paper, we motivate dDR as an alternative to iDR that enables direct translation between non-central domains. This is explicitly stated in lines 053-055 of the Introduction. The efficiency gains of dDR naturally come with a trade-off: its objective introduces a bias that can lead to a slight drop in quality relative to iDR, as we clearly explained in Section 4.2 (lines 212-214). Nonetheless, dDR still substantially outperforms existing baselines while requiring far fewer sampling steps. These results demonstrate that, despite the inherent bias, the learning strategy underlying dDR (Eq. 11) is *genuinely effective* and offers a compelling efficiency-quality balance compared to methods such as UniDiffuser.

---

> ### Author Response · Authors · 2025-11-23
> **Response to Reviewer AFk3 (2)**
>
> >**\[Weakness 2\]** "While dDR is computationally cheaper at inference, the paper does not adequately quantify this benefit against the measured drop in quality. The faster inference speed is valuable, but if it comes at the cost of a significant increase in FID score, iDR remains the superior method in terms of output quality. The paper would be much stronger if it provided a clear analysis of this trade-off, perhaps identifying scenarios where the speed of dDR is worth the quality sacrifice."
>
> **\[Answer\]** Our results in Tables 1$-$3 show that dDR offers substantial efficiency and effectiveness gains over existing baselines$-$requiring roughly *half the computation* while achieving *up to several-fold lower* FID scores compared to existing baselines, including the SoTA multi-domain translation method UniDiffuser. The performance gap between dDR and *our other proposed method*, iDR, is *small* (approximately 1$-$4 FID points) and *theoretically expected*. This difference is *far from "significant"*, as stated by Reviewer AFk3, and is well compensated by the considerable efficiency benefits of dDR. These characteristics make dDR particularly suitable in scenarios where inference cost is a key constraint. To more clearly illustrate the efficiency$-$quality trade-off between dDR and iDR, we compare their performance under the same number of sampling steps (NFE), ranging from 10 to 1000. The results are presented in the tables below:
> + FID score for non-cetral domains on **Faces-UMDT-Latent** w.r.t. different number of sampling step:
> | NFE | Ske.$\leftrightarrow$Seg. iDR | Ske.$\leftrightarrow$Seg. dDR |
> |:---:|:---:|:---:|
> | 10 | 74.33/7.95 | 39.29/6.65 |
> | 20 | 42.25/6.98 | 23.79/5.82 |
> | 30 | 31.77/6.67 | 22.31/5.72 |
> | 50 | 23.37/6.39 | 21.85/5.61 |
> | 100 | 18.05/6.25 | 20.82/5.58 |
> | 1000 | 16.17/6.19 | 19.42/5.52 |
>
> + FID score for non-cetral domains on **COCO-UMDT-Star** w.r.t. different number of sampling step:
>
> | NFE | Ske.$\leftrightarrow$Seg. iDR | Ske.$\leftrightarrow$Seg. dDR | Ske.$\leftrightarrow$Depth iDR | Ske.$\leftrightarrow$Depth dDR | Seg.$\leftrightarrow$Depth iDR | Seg.$\leftrightarrow$Depth dDR |
> |:---:|:---:|:---:|:---:|:---:|:---:|:---:|
> | 10 | 98.14/79.17 | 31.29/34.19 | 78.65/13.96 | 27.10/12.09 | 79.95/24.97 | 35.31/16.48 |
> | 20 | 50.09/32.07 | 28.26/25.67 | 42.95/11.30 | 22.69/11.26 | 32.93/15.35 | 26.03/15.63 |
> | 30 | 36.01/25.99 | 27.93/24.81 | 31.37/9.83 | 22.09/10.35 | 26.12/14.71 | 25.57/15.02 |
> | 50 | 27.84/23.45 | 27.19/24.06 | 19.39/9.03 | 21.65/9.59 | 24.87/13.93 | 25.21/14.74 |
> | 100 | 24.70/23.16 | 26.87/23.78 | 18.81/8.79 | 21.18/9.29 | 24.08/12.49 | 25.18/14.62 |
> | 1000 | 23.06/23.01 | 26.73/23.64 | 18.15/8.86 | 20.75/9.42 | 23.37/12.11 | 24.91/14.87 |
>
>
> It is evident that iDR's performance deteriorates sharply as NFE decreases, whereas dDR remains much more stable, with only a moderate drop in FID. For example, at NFE $\le$ 30, dDR achieves 50$-$100% lower FID scores than iDR on several translation tasks, such as Seg.$\rightarrow$Ske. on Faces-UMDT-Latent, and Ske.$\leftrightarrow$Seg., Ske.$\leftarrow$Depth, and Seg.$\leftarrow$Depth on COCO-UMDT-Star. This discrepancy arises because iDR relies on an intermediate central-domain sample whose quality degrades significantly when NFE is small, thereby impairing the final non-central-domain output. By contrast, dDR directly generates the non-central domain and thus avoids this issue.
>
> These results strongly indicate that dDR is not sensitive to the central domain sample quality and is the clearly preferable choice when sampling with a limited number of steps. We agree with Reviewer AFk3 that including this analysis strengthens the paper, and we will incorporate it into the revised version.
>
>
> > **\[Weakness 3\]** "The authors acknowledge that their tractable objective relies on a single-sample Monte Carlo estimate inside a logarithm, which introduces bias. This technical compromise is a likely source of dDR's underperformance. The work would be more convincing if it included analysis isolating the effect of this bias."
>
> **\[Answer\]** The single-sample Monte Carlo estimate inside the logarithm $\log\left(\mathbb{E}_{p\left({x'}^{c}|x^{i}\right)}\left[p\left(x^{j}|{x'}^{c}\right)\right]\right)$ is *critical* to deriving the *tractable* training objective for dDR in Eq. 8 and its scalable variational bound in Eq. 10. We agree with Reviewer AFk3 that isolating the effect of this bias would strengthen our work, and we genuinely attempted to explore this direction. However, mitigating this bias in a principled manner is *highly challenging* because doing so would *fundamentally alter* Eq. 8 into an intractable objective. In that case, we would require hundreds to thousands of diffusion denoising steps to sample $x'^{c}\sim p\left(x'^{c}\mid x^{i}\right)$, and we would also need to evaluate $p\left(x^{j}\mid x'^{c}\right)$, for which no closed-form density exists, as clearly stated in lines 200$-$204 of the paper.

---

### Author Response · Authors · 2025-11-23
**General Response**

Dear reviewers and AC,

We sincerely thank the reviewers for their time, effort, and careful evaluation of our manuscript. We are truly grateful that all four reviewers provided positive feedback, consistently highlighting several key strengths of our work:

- **Problem formulation (UMDT).** The formalization of Universal Multi-Domain Translation (UMDT) is described as "a significant contribution" that "addresses a critical limitation of existing multi-domain translation methods" and is "highly practical and forward-looking, with clear real-world applications" (AFk3), and as "well-motivated" with clearly explained background (es9u).

- **Diffusion Router (DR): idea + theory.** The core idea of Diffusion Router (DR) is called "an elegant and parameter-efficient solution" and iDR is said to be "a powerful model in its own right" (AFk3). The theoretical contribution for dDR is described as "clever" and "non-trivial" (AFk3). Using the central domain as a bridge and a single model parameterized by source/target domains is considered "reasonable and novel" (4zEu), and the method is summarized as "the proposed diffusion routers is novel" (ARK8).

- **Experiments and clarity.** The results are "promising" and that we "construct 3 different datasets and conduct extensive experiments" (ARK8); "extensive experiments validate the effectiveness of the proposed method" on star- and chain-structured tasks (es9u). The paper is also described as "easy to follow" with "clear" presentation and "clear" writing (4zEu, es9u).

As stated in the introduction, this work delivers four main contributions, which we recall here for better reference:

- We formally define UMDT, a general setting that aims to learn translations between any pairs of $K$ domains using only $K-1$ paired datasets sharing a central domain.

- We introduce DR, a single diffusion-based framework that models all central$\leftrightarrow$non-central mappings with a single noise predictor.

- We develop a scalable variational-bound objective together with Tweedie refinement to enable direct non-central translations without requiring training pairs.

- We construct three new UMDT benchmarks and show that DR achieves state-of-the-art performance on both indirect and direct translations across different UMDT topologies.

The key message of this paper is therefore to establish UMDT as a well-motivated, tractable problem, to show that DR is a scalable and effective solution for universal translation under realistic constraints, and to provide three new benchmarks that enable controlled, reproducible evaluation of these ideas.

While extending DR to fully multi-modal settings (e.g., image$-$text$-$audio) is a natural and exciting next step, thoroughly addressing this would require non-trivial modality-specific design and large-scale compute, and would distract from our key message. As stated in the Conclusion, we therefore treat multi-modal UMDT as separate, planned future work, and we are actively pursuing these extensions as a strong, standalone follow-up study.

---

> ### Author Response · Authors · 2025-12-02
> **Summary of Revisions**
>
> Dear Reviewers and AC,
>
> In the revised version of the manuscript, we have updated the paper in response to the reviewers’ comments. The modified parts in the updated paper are highlighted in $\textcolor{blue}{blue}$. The main changes are:
>
> + Added a comparison between iDR and dDR across different numbers of sampling steps in Appendix D.3.4 (Reviewer AFk3).
>
> + Added a sentence in Sec. 5.1.2 to explicitly clarify the X/Y format used in the result tables (Reviewer es9u).
>
> Sincerely,
>
> The Authors

---

### Meta-Review · Area_Chair_xfeK · 2026-01-02

**Summary:**

This paper proposes Universal Multi-Domain Translation (UMDT), a novel problem setting that enables translation between any pair of K domains using only K-1 paired datasets with a central domain. To address this problem, the authors introduce Diffusion Router (DR), a unified diffusion-based framework comprising iDR and dDR via variational-bound objective and Tweedie refinement. The work also constructs three large-scale UMDT benchmarks and demonstrates state-of-the-art performance. All four reviewers recognized the paper’s strengths, and the authors’ rebuttals have addressed core concerns. Thus, the paper meets ICLR 2026 acceptance criteria, leading to an Accept decision.

**Reviewer Scores:**

NA

---

### Decision · Program_Chairs · 2026-01-26

Accept (Poster)